# An Efficient Training Algorithm for Models with Block-wise Sparsity

**Ding Zhu**                                                          *zhu.3723@osu.edu*
*Department of Computer Science and Engineering*
*The Ohio State University*

**Zhiqun Zuo**                                                        *zuo.167@osu.edu*
*Department of Computer Science and Engineering*
*The Ohio State University*

**Mohammad Mahdi Khalili**                                           *khalili.17@osu.edu*
*Department of Computer Science and Engineering*
*The Ohio State University*

**Reviewed on OpenReview:** *https://openreview.net/forum?id=nay3Kvw8BD*

## Abstract

Large-scale machine learning (ML) models are increasingly being used in critical domains like education, lending, recruitment, healthcare, criminal justice, etc. However, the training, deployment, and utilization of these models demand substantial computational resources. To decrease computation and memory costs, machine learning models with sparse weight matrices are widely used in the literature. Among sparse models, those with special sparse structures (e.g., models with block-wise sparse weight matrices) fit better with the hardware accelerators and can decrease the memory and computation costs during the inference. Unfortunately, while there are several efficient training methods, none of them are designed to train a block-wise sparse model efficiently. As a result, the current methods for training block-wise sparse models start with full and dense models leading to inefficient training. In this work, we focus on training models with *block-wise sparse matrices* and propose an efficient training algorithm to decrease both computation and memory costs during training and inference. In addition, we will show that our proposed method enables us to efficiently find the right block size for the sparsity pattern during the training process. Our extensive empirical and theoretical analyses show that our algorithms can decrease the computation and memory costs significantly without a performance drop compared to baselines.[1]

## 1 Introduction

Deep learning models have achieved remarkable success across various domains, but training these models on resource-constrained devices remains challenging due to their computational and memory requirements. To decrease memory and computational costs, sparse machine learning models have been widely used in literature. Sparse networks are mainly categorized into two groups: fine-grained sparse networks and coarse-grained sparse networks (see Figure 1 which provides an example of a fine-grained sparse weight matrix and two examples of coarse-grained sparse weight matrices). In fine-grained sparse networks, the weight matrices are sparse but they do not have any special structure. This type of sparsity generally improves the storage cost but does not improve the inference time significantly. This is because the random distribution in fine-grained sparse weight matrices does not fit with the hardware accelerators, and they can speed up

---

[1]A preliminary version of this work was presented at the NeurIPS Workshop on Machine Learning and Compression (Zhu et al., 2024).

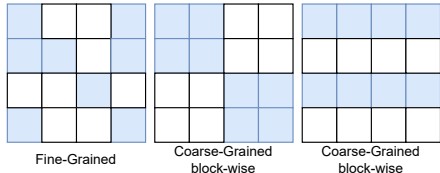

Figure 1: Examples for fine-grained and coarse-grained sparse matrices. White entries represent zero value.

the inference time only if the sparsity ratio is higher than 95% (Wang et al., 2018; Wen et al., 2016). On the other hand, coarse-grained sparse matrices are better alternatives to speed up inference (Parashar et al., 2017; Han et al., 2017; 2016a).

To train coarse-grained (structured) sparse weight matrices, it is common to use iterative pruning (Tan & Motani, 2020; Yu et al., 2023) or group lasso techniques (Behnke & Heafield, 2021; Rao et al., 2015; Ida et al., 2019). Iterative pruning works as follows. First, a sparsity pattern is defined (e.g., 2:4 sparsity (Mishra et al., 2021) or channel-wise sparsity (Liu et al., 2017)). Based on this pattern, the weight parameters are divided into separate groups. A deep model is trained and then by looking at weight matrices, we prune some of the groups that are not impacting the performance of the model. Then, we re-train (fine-tune) the remaining weights and again prune those groups that have the smallest impact on the performance. This procedure is repeated until we achieve the desired sparse network. Group LASSO technique adds a regularizer to the objective function which ensures that weights in several groups go to zero during the training. In particular, for Group LASSO, first, we divide the parameters in each weight matrix into several groups (each block can be considered as a group) and add a regularizer for each group to the objective function. As a result, the weights in certain groups with minimal impact on accuracy will eventually go to zero during training. Nevertheless, iterative pruning and group lasso techniques mainly focus on reducing memory and computation costs at the time of inference. However, they can be very costly during the training as the training process starts with the full network (i.e., all the weights are non-zero at the beginning) and the gradient needs to be calculated with respect to all the model parameters.

In this paper, we propose a technique that enables us to train block-wise sparse matrices that improve both computation and memory costs during training and inference. We leverage the Kronecker product decomposition to propose a new matrix factorization technique suitable for block-wise sparse matrices. Our contributions in this work can be summarized as follows,

- To the best of our knowledge, this is the first work that focuses on efficient training for block-wise sparse models. This work is also the first work that makes a connection between block-wise sparse matrices and the Kronecker product decomposition for efficient training. While there have been several algorithms for training block-wise sparse models (Cai et al., 2022; Liu et al., 2021a; Li et al., 2017a; Oyedotun et al., 2020a), they are not efficient during training.

- We theoretically demonstrate every block-wise sparse matrix can be represented using our proposed decomposition resulting in no significant performance drop.

- Through extensive theoretical analyses, we show that our proposed approach decreases the number of flops and training parameters during the training compared to existing methods for training block-wise sparse models.

- Through extensive empirical study, we show that in some cases, our proposed method can reduce the number of training parameters and training FLOPs by 97% with a minimal accuracy drop.

- We also show that our proposed algorithm is able to find the optimal block size for block-wise sparse matrices efficiently in one round of training.

The remainder of the paper is organized as follows. We review related work in Section 2. We present the preliminaries in Section 3, followed by problem formulation in Section 4. We discuss how our algorithm can

be used for optimal pattern selection in Section 5. We present numerical results in Section 6, and conclude in Section 7.

## 2 Related Work

**Pruning** is an effective technique for reducing model parameters and has seen tremendous progress in recent years (Reed, 1993; Tanaka et al., 2020b; Han et al., 2016b). Pruning generally can be divided into structured pruning (Cai et al., 2022; Liu et al., 2021a; Li et al., 2017a) and unstructured pruning (Dong et al., 2017; Sanh et al., 2020). Unstructured pruning finds fine-grained sparse matrices by setting the unimportant weights to zero (See Figure 1). While unstructured pruning can decrease the number of model parameters at the inference time, it generally does not improve the inference time (Wang et al., 2018). Structured pruning, on the other hand, trains coarse-grained sparse matrices leading to a decrease in the memory cost and inference time. It is important to note that pruning methods generally start with a dense full network and prune the network in one-shot or in several iterations and retrain the network to improve the performance. As a result, memory and computation costs during the training can be expensive. Recently, several pruning methods have been proposed to decrease the training cost as well. These methods try to prune the network right after initialization and train the sparse network (Lee et al., 2019a). The current Pruning After Initialization (PAI) methods are able to perform unstructured pruning, and they need large memory at the time of initialization which makes it impossible to do training on small devices. On the other hand, our proposed method is able to decrease the number of parameters and flops from the beginning of the initialization and train block-wise sparse matrices to decrease inference memory and time.

**Regularization** is another method for training a network with sparse weight matrices. It is common to add l1 or l0 regularizer to the loss function (Ma et al., 2019; Louizos et al., 2017) to find unstructured sparse weight matrices. Group LASSO, an extension of the LASSO method, is a method that imposes a regularizer to pre-defined groups of model parameters leading to block-wise or group-wise sparsity structures in deep neural networks (Liu et al., 2015; Scardapane et al., 2017; Behnke & Heafield, 2021). To improve the performance of group LASSO, recently a new variation of group LASSO called elastic group LASSO (Oyedotun et al., 2020a) has been proposed. It is worth mentioning that group LASSO is only able to remove the computation and memory cost during inference. The training cost associated with group LASSO is relatively high as this method starts the training with a dense network.

**Matrix/tensor factorization** (Yu et al., 2017; Wu & Feng, 2022) is a compression method that is able to reduce the training and inference cost by reducing the number of training parameters from the beginning of the training process. While matrix/tensor factorization has been widely used for model compression (Hsu et al., 2022; Hameed et al., 2022; Edalati et al., 2021; Yin et al., 2022), these methods are not able to find block-wise sparse matrices for deep models.

**Knowledge Distillation** (KD) methods train a smaller student network based on the guidance of a bigger teacher network (Hinton et al., 2015; Li et al., 2017b). Knowledge distillation methods try to make sure that the student network mimics the behavior of teacher network by comparing outputs (Cho & Hariharan, 2019; Furlanello et al., 2018; Mirzadeh et al., 2020; Zhang et al., 2018) or intermediate features (Heo et al., 2019a;b; Huang & Wang, 2017; Kim et al., 2018; Park et al., 2019; Tian et al., 2019). The training process under KD can be computationally heavy as we need to train a teacher model first and then use the teacher model to train a student model (Yim et al., 2017).

**Block-wise Sparsity** is a special type of structured sparsity. Zero elements of a block-wise sparse matrix can be stored contiguously in memory reducing irregular memory access and taking advantage of array-data-path in modern processors. D'Alberto et al. (2024) show that block-wise sparse models can run efficiently on specific devices. In the literature, block-wise sparsity has been implemented by using LASSO regulation (Narang et al., 2017), group regulation (Wen et al., 2016; Vooturi et al., 2018), and filtering (Anwar et al., 2015).

**Sparse Training** aims to decrease number of parameters that are being updated during gradient descent. Sparse training methods generally are categorized into two groups: static sparse training and dynamic sparse training. Static sparse training (Dao et al., 2022; Lee et al., 2019b; Yuan et al., 2021) identifies important

parameters and finds the sparse mask at beginning of training based on a heuristic metric. As a result, only a subset of parameters are updated during the gradient descent. For example, Grasp method (Wang et al., 2020) uses gradient flow, SynFlow method (Tanaka et al., 2020a) uses synaptic strengths, and FISH mask (Sung et al., 2021) uses Fisher information to find the sparse mask. Dynamic sparse training, on the other hand, allows to modify the sparsity pattern during training to reduce the training footprint. For example, Mocanu et al. (2018) use the weight magnitude to prune the connection between neurons during the training, and RigL (Evci et al., 2020) uses both weight magnitude and gradient magnitude to identify important weights in each epoch and update them. However, these efficient training methods are working with unstructured sparse mask and may not lead to training acceleration. Fang et al. (2024) consider N:M sparsity pattern for hardware acceleration and implement an efficient training for this sparsity pattern. Jiang et al. (2024) also propose an efficient training scheme by shuffling rows or column based on the Jaccard similarity. While these approaches can reduce the number of training parameters with structured sparsity, they are designed to find block-wise sparse models.

In this paper, our goal is to propose a new method for training block-wise sparse model that is efficient during both training and inference. The proposed method can decrease the number of training parameters and flops significantly without degrading the model performance compared to the baselines.

## 3 Preliminary

**Group LASSO:** LASSO is a method for learning sparse models by adding a penalty term to the cost function, which is proportional to the $l_1$ norm of the model's coefficients. This encourages the model to be sparse and sets some of the model parameters to be zero. Group LASSO is an extension of LASSO which divides the model parameters into several groups and imposes a regularizer on each group. This encourages the coefficient in several groups to go to zero during the training. Group LASSO can be used to train block-wise sparse matrices (see Figure 2) by defining each block as a group. In particular, consider a neural network with $L$ layers, with $\mathbf{W}$ being the weight matrices of the whole network and $W^{[l]}$ being the weight matrix in layer $l$. Also, let $W_g^{[l]}$ denotes the block/group $g$ in layer $l$. Then, the group LASSO solves the following optimization problem,

$$\hat{\mathbf{W}}_\lambda = \arg\min_{\mathbf{W}} \mathcal{J}(W^{[1]}, \dots, W^{[L]}; \mathcal{D}) + \lambda \sum_{l=1}^{L} \sum_{g} ||W_g^{[l]}||_F, \tag{1}$$

where $\hat{\mathbf{W}}_\lambda$ is the optimized weight matrices with hyperparameter $\lambda$, $\mathcal{J}$ is the loss function, and $\mathcal{D}$ is the training dataset, and $||.||_F$ denotes the Frobenius norm.

**Kronecker Product Decomposition:** The Kronecker product, denoted by $\otimes$, is a mathematical operation that combines two matrices to form a larger matrix. Given two matrices $A$ and $B$, if $A$ is of size $m_1 \times n_1$ and $B$ is of size $m_2 \times n_2$, then the Kronecker product of A and B results in a matrix of size $m_1 m_2 \times n_1 n_2$. Let $W$ be an $m$ by $n$ matrix, where $m = m_1 m_2, n = n_1 n_2$ we can decompose this matrix using the Kronecker product decomposition as follows (Van Loan, 2000),

$$W = \sum_{i=1}^{R} A_i \otimes B_i = \sum_{i=1}^{R} W_i, \tag{2}$$

where $A_i$ is an $m_1$ by $n_1$ matrix, $B_i$ is an $m_2$ by $n_2$ matrix, and $R = \min\{m_1 n_1, m_2 n_2\}$. Given this decomposition, $Wx$ (where $x \in \mathbb{R}^{n \times 1}$) also can be calculated as follows, $Wx = vec(\sum_{i=1}^{R} B_i \check{x} A_i^T)$, where $\check{x}$ is $n_2$ by $n_1$ matrix and can be obtained by re-arranging the elements of vector $x$ (Van Loan, 2000).

It turns out that the low-rank approximation is a special case of decomposition equation 2. More precisely, if we set $m_2 = 1$ and $n_1 = 1$, Equation 2 is equivalent to the low-rank approximation. Let's assume that we want to approximate matrix $W$ by $r$ terms of equation 2. In this case, $W \approx W_r = \sum_{i=1}^{r} A_i \otimes B_i$, where $W_r$ is expressed by $r(m_1 n_1 + m_2 n_2)$ parameters.

It is worth mentioning that similar to the low-rank approximation, Kronecker product decomposition can be used for compressing deep models (Jagtap et al., 2022; Hameed et al., 2022). However, to the best of our

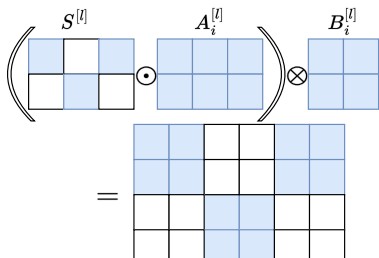

Figure 2: Illustration of why equation 3 leads to block-wise sparsity when $S^{[l]}$ is sparse. White entries represent zero value.

knowledge, there is no tensor/matrix factorization for generating and training block-wise sparse matrices. In the next section, we will explain how we can take advantage of Kronecker product decomposition to train efficiently block-wise sparse matrices and reduce the memory footprint during the training process.

## 4 Problem Statement and Proposed Solution

As we discussed in Section 2, there are several methods including iterative pruning or group LASSO to train block-wise sparse weight matrices. However, these methods have to start with a full uncompressed model and sparsify the weight matrices gradually during the training. As a result, these methods do not decrease computation and memory during training. However, in this part, we propose a new matrix decomposition method leveraging the Kronecker product decomposition algorithm to train block-wise sparse matrices with fewer training parameters and fewer flops for forward and backward propagation compared to the group LASSO and structured pruning approaches during training. During the training period, we utilize decomposed matrices, which reduces the number of training parameters and flops. This allows the training process to be deployed on resource constrained devices. During inference, our algorithm directly uses block-wise sparse matrices without any decomposition to improve efficiency.

To demonstrate how to learn a structured sparse weight matrix efficiently, assume that $W^{[l]}$ is the weight matrix associated with layer $l$ (if we are working with a convolutional neural network, $W^{[l]}$ can be a tensor.) Instead of learning $W^{[l]}$, we propose to estimate it by

$$W_r^{[l]} = \sum_{i=1}^{r_l} (S^{[l]} \odot A_i^{[l]}) \otimes B_i^{[l]}, \tag{3}$$

where $r_l$ is a hyper-parameter called rank, $\odot$ is element-wise product, $S^{[l]}$ and $A_i^{[l]}$ are $m_1$ by $n_1$ matrices and $B_i^{[l]}$ is an $m_2$ by $n_2$ matrices. Then, we can train $S^{[l]}, (A_i^{[l]}, B_i^{[l]})_{i=1}^{r_l}$ directly during the training process (we calculate the gradient of the loss function with respect to these parameters). By imposing an $l_1$ regularizer on $S^{[l]}$, we can make sure that $S^{[l]}$ is sparse in the following problem,

$$\min_{[S^{[l]},A_i^{[l]},B_i^{[l]}]_{i \leq r_l, l \leq L}} \mathcal{J}([S^{[l]}, A_i^{[l]}, B_i^{[l]}]_{i \leq r_l, l \leq L}, \mathcal{D}) + \lambda \sum_{l=1}^{L} ||S^{[l]}||_1, \tag{4}$$

where $\lambda$ is a constant and controls the sparsity rate. If $S^{[l]}$ is an unstructured sparse matrix, $W_r^{[l]} = \sum_{i=1}^{r_l} (S^{[l]} \odot A_i^{[l]}) \otimes B_i^{[l]}$ will be a block-wise sparse matrix (see Figure 2). After training, depending our application, we can use $W_r^{[l]}$ during the inference time or we can use $S^{[l]}, (A_i^{[l]}, B_i^{[l]})_{i=1}^{r_l}$ directly during the inference time. It is worth mentioning that the decomposition in equation 3 provides several degrees of freedom and hyper-parameters (i.e., $m_1, m_2, n_1, n_2, r_l$). Note that the block size in $W_r^{[l]}$ is determined by the size of matrix $B_i^{[l]}$ (i.e., $(m_2, n_2)$). To choose the right hyper-parameters, it is common to choose to try different sets of hyper-parameters and pick the one with desirable performance and efficiency. If our only goal is to minimize the number of parameters using equation 3, then the hyper-paramters can be determined by an optimization problem. In particular, we can set $r_l = 1$ and solve the following integer programming,

$$\min_{m_1, n_1, m_2, n_2} 2m_1 n_1 + m_2 n_2, s.t., m_1 m_2 = m, n_1 n_2 = n, \tag{5}$$

where the objective function is equal to the number of parameters in equation 3. The above optimization problem is nonconvex. We can solve it by setting $m_2 = m/m_1$ and $n_2 = n/n_1$. The above optimization problem reduces to $\min_{m_1, n_1} 2m_1 n_1 + \frac{mn}{m_1 \cdot n_1}$. By the first order condition, the minimizer of $2m_1 n_1 + \frac{mn}{m_1 \cdot n_1}$ is $m_1 n_1 = \sqrt{0.5 \cdot mn}$ if $\sqrt{0.5 \cdot mn}$ is integer (if $\sqrt{0.5 \cdot mn}$ is not an integer, we can use integer programming tools like branch and bound to solve the problem). To clarify, we provide an example.

**Example 1.** *Let $m = 2^3$ and $n = 2^8$. In this case, for the minimum possible space complexity under factorization equation 3, we need to have $m_1 n_1 = 32$. For example, we can set $m_1 = 4, n_1 = 8, m_2 = 2, n_2 = 32$. In this case, the total number of parameters would be $128$. The original matrix $W^{[l]}$, has $2048$ training parameters. Therefore, at the time of training, using equation 3, we need to train $128$ variables while the group LASSO technique needs to train $2048$ variables. At the inference time, we can use directly sparse matrix $(S^{[l]} \odot A_1^{[l]})$ and $B_1^{[l]}$ to make an inference. We can also use block-wise sparse matrix $W_1^{[l]}$ to make an inference.*

The above example shows the proposed factorization in equation 3 can reduce the training parameters significantly. We can also optimize $r_l, m_1, n_1, m_2, n_2$ for maximizing accuracy and performance. We will discuss how we can choose $r_l, m_1, n_1, m_2, n_2$, and the right block size for better accuracy in Section 5.

In the remaining part of this section, we explain how any block-wise sparse matrix can be represented by equation 3 and why the proposed factorization can decrease the number of flops during forward and backward propagation compared to group LASSO/Structured Pruning.

**Proposition 1.** *Let $\hat{W}^{[l]}$ be a block-wise sparse matrix trained by group LASSO or iterative pruning. If the blocks have the same size, then there exists $\hat{r}_l$ and $\hat{S}^{[l]}$ and $(\hat{A}_i^{[l]}, \hat{B}_i^{[l]})_{i=1}^{r_l}$ such that $\hat{W}^{[l]} = \sum_{i=1}^{\hat{r}_l} (\hat{S}^{[l]} \odot \hat{A}_i^{[l]}) \otimes \hat{B}_i^{[l]}$.*

Intuitively, the above proposition implies that if the hyper-parameters $r_l, n_1, n_2, m_1, m_2$ are chosen correctly, then training matrices $S^{[l]}, (A_i^{[l]}, B_i^{[l]})_{i=1}^{r_l}$ should have the same performance of a model trained by the group LASSO or pruning technique.

*Proof.* Here, we will prove that each block-wise sparse matrix can be decomposed by the decomposition in equation 3.

We assume the block-wise sparse matrix $\hat{W}^{[l]}$ has a size of $m$ by $n$, the number of groups is $m_1 \times n_1$ and the block size is $m_2$ by $n_2$ where $m_1 \times m_2 = m$ and $n_1 \times n_2 = n$. We want to find $\hat{S}^{[l]}, \hat{A}_i^{[l]}, \hat{B}_i^{[l]}$ such that the following holds,

$$\hat{W}^{[l]} = \sum_{i=1}^{r} (\hat{S}^{[l]} \odot \hat{A}_i^{[l]}) \otimes \hat{B}_i^{[l]}, \tag{6}$$

where $\hat{W}^{[l]}$ is given.

Assume that $T$ groups among $n_1 m_1$ groups are non-zero in matrix $\hat{W}^{[l]}$ and the index of the groups are $t_1, t_2, ..., t_T$. We set $r = T$ and we generate a series of $\hat{A}_i^{[l]}$ and $\hat{B}_i^{[l]}$ for $i = 1, ..., T$ to make sure equation 6 holds. In particular, we set $\hat{B}_i^{[l]}$ matrix to be equal to the block of $t_i$ in matrix $\hat{W}^{[l]}$, and $\hat{A}_i^{[l]}$ is a matrix that only has one entry equal to 1 which is associated with block $t_i$ and the other entries are zeros. We also can set $\hat{S}^{[l]}$ to be a binary matrix. Each entry in $\hat{S}^{[l]}$ corresponds to a block in $\hat{W}^{[l]}$. If a block in $\hat{W}^{[l]}$ is non-zero (resp. zero), then the entry associated with that block in $\hat{S}^{[l]}$ would be 1 (resp. 0). By this construction, $\hat{S}^{[l]}, \hat{A}_i^{[l]}, \hat{B}_i^{[l]}$ satisfy equation 6. $\qquad \square$

**Proposition 2.** *[Number of Flops for Forward and Backward Passes in A Linear Model] Consider multivariate linear regression model $h(x) = Wx$, where $W$ is an $m$ by $n$ matrix, and $x \in \mathbb{R}^n$ is the input feature vector. Let $\mathcal{D} = \{(x_j, y_j)|j = 1, ..., N\}$ be the training dataset, and $\mathcal{J}(W; \mathcal{D}) = \sum_{j=1}^{N} ||Wx_j - y_j||_2^2$ be the objective function. If we estimate $W$ by $\sum_{i=1}^{r} (S \odot A_i) \otimes B_i$ and write $\mathcal{J}(S, (A_i, B_i)_{i=1}^{r}; \mathcal{D}) = \sum_{j=1}^{N} ||\sum_{i=1}^{r} [(S \odot A_i) \otimes B_i]x_j - y_j||_2^2$, then,*

- **Forward pass:** *Number of flops for calculating $\mathcal{J}(W; \mathcal{D})$ is $\mathcal{O}(2(n+1)mN)$. On the other hand, $\mathcal{J}(S, (A_i, B_i)_{i=1}^r; \mathcal{D})$ can be calculated by $\mathcal{O}\left(2Nrm_1n_1(m_2+n_2) - Nr(m+2m_2n_1) + 3Nm\right)$ flops.*

- **Backward pass:** *Number of flops for calculating gradient of $\mathcal{J}(W; \mathcal{D})$ with respect to $W$ is $\mathcal{O}(mN(2n+1))$. On the other hand, $\frac{\partial \mathcal{J}(S, (A_i, B_i)_{i=1}^r; \mathcal{D})}{\partial A_i}$ and $\frac{\partial \mathcal{J}(S, (A_i, B_i)_{i=1}^r; \mathcal{D})}{\partial B_i}$ can be calculated by $\mathcal{O}(2Nn_1m_2(n_2+m_1) - Nn_1m_2 + m_1n_1)$ flops and $\mathcal{O}(2Nm_1n_2(m_2+n_1) + m_1n_1 - Nn_2m_1)$ flops, respectively. In addition, $\frac{\partial \mathcal{J}(S, (A_i, B_i)_{i=1}^r; \mathcal{D})}{\partial S}$ needs $\mathcal{O}\left(Nm + Nr(4m_1m_2n_1 - m_2n_1 + 2m_2n_1n_2)\right)$ flops.*

The above proposition implies forward and backward passes for $\mathcal{J}(S, (A_i, B_i)_{i=1}^r, \mathcal{D})$ can be more efficient compared to calculating $\mathcal{J}(W, \mathcal{D})$ if right values for parameters $m_1, n_1, m_2, n_2, r$ are chosen. This is because the forward and backward passes for $\mathcal{J}(W, \mathcal{D})$ needs $\mathcal{O}(mnN)$ while $mn$ does not appear in the time complexity of forward and backward passes of $\mathcal{J}(S, (A_i, B_i)_{i=1}^r, \mathcal{D})$. In addition to flops for calculating the gradient, in each epoch of gradient descent for $\mathcal{J}(W, \mathcal{D})$, we need to update $mn$ parameters which needs $\mathcal{O}(mn)$ flops. On the other hand, in each epoch of gradient descent for $\mathcal{J}(S, (A_i, B_i)_{i=1}^r, \mathcal{D})$, after calculating the gradient, we update $r(m_1n_1 + m_2n_2)$ parameters which need $\mathcal{O}(r(m_1n_1 + m_2n_2))$ flops. This is another reason that the training process can be more efficient under decomposition equation 3.

We want to emphasize that our experimental results also show that number flops during the training can be significantly decreased by decomposition equation 3 (e.g., Table 1 shows 87% reduction in number flops without any accuracy drop, and Table 3 shows 97% reduction in number flops with 1 percentage point performance drop compared to the dense model.) Next, we extend the results of Proposition 2 to a network with two layers. We will also perform analysis for a larger network in Section 6.

**Proposition 3.** *Consider a two-layer multivariate regression model $h(x) = W^{[2]}\sigma(W^{[1]}x)$, where $W \in \mathbb{R}^{m^{[2]} \times m^{[1]}}$ is the weight matrix of the first layer, and $W^{[2]} \in \mathbb{R}^{m^{[3]} \times m^{[1]}}$ is the weight matrix of the second layer. $x \in \mathbb{R}^{m^{[1]}}$ is the input feature vector. Let $\mathcal{D} = \{(x_j, y_j) | j = 1, \ldots, N\}$ be the training dataset, and $\mathcal{J}(W^{[1]}, W^{[2]}; \mathcal{D}) = \sum_{j=1}^N \|W^{[2]}\sigma(W^{[1]}x_j) - y_j\|_F^2$ be the objective function. Assume that we estimate $W^{[1]}$ by $\sum_{i=1}^{r^{[1]}} (S^{[1]} \odot A_i^{[1]}) \otimes B_i^{[1]}$ and $W^{[2]}$ by $\sum_{i=1}^{r^{[2]}} (S^{[2]} \odot A_i^{[2]}) \otimes B_i^{[2]}$. In the first layer, $S^{[1]}$ and $A_i^{[1]} \in \mathbb{R}^{m_1^{[2]} \times m_1^{[1]}}$, $B_i^{[1]} \in \mathbb{R}^{m_2^{[2]} \times m_2^{[1]}}$, where $m_1^{[1]}m_2^{[1]} = m^{[1]}$, $m_1^{[2]}m_2^{[2]} = m^{[2]}$. In the second layer, $S^{[2]}$ and $A_i^{[2]} \in \mathbb{R}^{m_1^{[3]} \times m_1^{[2]}}$, $B_i^{[2]} \in \mathbb{R}^{m_2^{[3]} \times m_2^{[2]}}$, where $m_1^{[3]}m_2^{[3]} = m^{[3]}$. Then,*

- **Forward pass:** *Number of flops for calculating $\mathcal{J}(W^{[1]}, W^{[2]}; \mathcal{D})$ is $\mathcal{O}\left(2N(m^{[1]}m^{[2]} + m^{[2]}m^{[3]} + m^{[3]})\right)$. Number of flops with the decomposition in equation 3 is $\mathcal{O}\left(r^{[1]}C_1 + r^{[2]}C_2 + Nm^{[2]} + 3Nm^{[3]}\right)$, where $C_1 = 2Nm^{[1]}m_2^{[2]} + 2Nm^{[2]}m_1^{[1]} - Nm_2^{[2]}(m_1^{[1]} + m_1^{[2]})$, $C_2 = 2Nm^{[2]}m_2^{[3]} + 2Nm^{[3]}m_1^{[2]} - Nm_2^{[3]}(m_1^{[2]} + m_1^{[3]})$.*

- **Backward pass:** *Number of flops for calculating gradients with respect to $W^{[1]}$ and $W^{[2]}$ is $\mathcal{O}\left(N(2m^{[1]}m^{[2]} + 4m^{[2]}m^{[3]} + m^{[3]})\right)$. Number of flops with the decomposition in equation 3 is $\mathcal{O}\left(N(m^{[2]} + m^{[3]}) + C_3 + C_4\right)$, where $C_3 = r^{[2]}Nm_1^{[2]}(4m^{[3]} - m_2^{[3]}) + 2r^{[2]}Nm^{[2]}m_2^{[3]}$, $C_4 = r^{[1]}Nm_1^{[3]}(4m^{[2]} - m_2^{[2]}) + 2r^{[1]}Nm^{[1]}m_2^{[2]}$.*

The above proposition implies that the results of Proposition 2 can be extended to non-linear models. In particular, Proposition 3 implies that if ranks $r^{[1]}$ and $r^{[2]}$ are small enough and right dimensions for $A_i^{[1]}, A_i^{[2]}, B_i^{[1]}, B_i^{[2]}$ are selected, then the decomposition in equation 3 can make the forward and backward passes more efficient in a two-layer neural network. This is because $m^{[1]} \cdot m^{[2]}$ and $m^{[2]} \cdot m^{[3]}$ do not appear in the time complexity after the decomposition.

While we solve the proposed optimization problem 4 during training, at the inference time, we use sparse matrix $\hat{W}^{[l]} = \sum_{i=1}^r (\hat{S}^{[l]} \odot \hat{A}_i^{[l]}) \otimes \hat{B}_i^{[l]}$. As a result, during inference, the memory and flops reduction for the proposed method compared to the the dense model is approximately proportional to the sparsity rate (note that actual reduction depends on the implementation of sparse matrices in pytorch, tensorflow, or other deep learning libraries, and sparsity rate is only a proxy for reduction in memory and flops during inference.). During training, the memory required for model parameters is proportional to the number of training parameters in optimization problem 4.

## 5 Pattern Selection for Performance Improvement

As we discussed in the previous section, the block size for training block-wise sparse matrices is a hyper-parameter, and selecting the right block size depends on our objective. For instance, by solving equation 5, we can find the block size that minimizes the number of training parameters. In this section, we explain how we can find the right block size for maximizing accuracy with only one round of training. We will argue that under factorization stated in equation 3, we can efficiently find the right block size compared to group LASSO and iterative pruning. First, we want to emphasize the number of possible block sizes for training block-wise sparse matrices is finite. For example, if the size of matrix $W^{[l]}$ is 10 by 10, then there are 14 possible block sizes: 1 by 10, 1 by 5, 1 by 2, 2 by 10, 2 by 5, 2 by 2, 2 by 1, 5 by 10, 5 by 5, 5 by 2, 5 by 10, 10 by 1, 10 by 2, 10 by 5. If we want to consider a block size with an exponent of 2, then 2 by 2 will be the only option. Let $P = \{P_1, P_2, ..., P_K\}$ be a set of finite sparsity patterns that we are interested in.[2] Each $P_k$ determines the block size for all the layers. If we want to use iterative pruning or group LASSO technique to find the best pattern, we have to run the training procedure for each $P_k$ and pick the one that achieves the highest accuracy. On the other, our method can select a pattern leading to the best accuracy in one round of training while keeping the number of training parameters less than that of group LASSO or iterative pruning. In particular, we estimate $W^{[l],(k)} = \sum_{i=1}^{r_l} [S^{[l],(k)} \odot A_i^{[l],(k)}] \otimes B_i^{[l],(k)}$, where $S^{[l],(k)}, A_i^{[l],(k)}, B_i^{[l],(k)}, 1 \le l \le L$ follows pattern $P_k$. Then, for the pattern selection, we solve the following optimization problem,

$$\min \sum_{k=1}^{K} J([S^{[l],(k)}, A_i^{[l],(k)}, B_i^{[l],(k)}]_{i \le r_l, l \le L}, \mathcal{D}) + \lambda_1 \sum_{k}^{K} \sqrt{\sum_{l=1}^{L} ||S^{[l],(k)}||_F^2} + \lambda_2 \sum_{i=1}^{K} \sum_{l=1}^{L} ||S^{[l],(k)}||_1, \tag{7}$$

where $[S^{[l],(k)}, A_i^{[l],(k)}, B_i^{[l],(k)}]_{i \le r_l, l \le L}$ are the training parameters, and $\lambda_1$ and $\lambda_2$ are regularizer parameters. The above optimization problem can be solved using gradient descent. The second term in equation 7 imposes a separate regularizer on each pattern. Since the square root is not differentiable at zero, this type of regularizer encourages all the weight associated with a pattern to go to zero if the pattern does not provide high performance (similar to the group LASSO technique). In the training process, we would gradually increase $\lambda_1$ until only one group of parameters remains non-zero. The third term encourages the trained $S^{[l](k)}$ to be sparse. If matrix $S^{[l],(k)}$ is sparse, then the weight matrices in later $l$ are block-wise sparse.

The benefit of using optimization problem equation 7 for pattern selection is that the number of training parameters significantly reduced compared to group LASSO or iterative pruning. Also, compared with the iterative pruning or group LASSO, we only need one round of training to get the right pattern structure. On the other hand, under group LASSO or iterative pruning, we need to run the training process for each pattern separately leading to heavy computation costs. To make this clear, we provide an example.

**Example 2.** *Consider a linear model with a weight matrix with dimension $(m = 2^3, n = 2^8)$. Assume that we have $K = 2$ options for the block-size: 4 by 4 and 8 by 8. Let fix $r_l = 4$. In this case, for optimization problem equation 7, we have $5 \times (2 \times 2^6 + 2^5) + 4 \times (4 \times 4 + 8 \times 8) = 1120$ parameters for pattern selection. On the other hand, the group LASSO technique needs to train $2048$ variables for each pattern.*

## 6 Experiment

### 6.1 Linear Model on MNIST

The MNIST dataset contains 60000 training images and 10000 testing images which are gray pictures of digits 0 to 9. The size of each image is $28 \times 28$ pixels. In the first experiment, we train a linear layer with a softmax activation function that takes a flattened image as its input.

---

[2]Generally, we need to consider a handful of options for block size. For instance, it is common to use the same block size across different layers and choose a block size that is an exponent of 2 (Gray et al., 2017). In our experiment, we consider $K \le 5$.

Table 1 compares our algorithm which is based on KDP with group LASSO (Scardapane et al., 2017) and the elastic group LASSO (Oyedotun et al., 2020b), and RigL (Evci et al., 2020). We run our algorithm, group LASSO, elastic group LASSO, and RigL for different block sizes. Note that RigL algorithm has not been designed for training block-wise sparse models. However, we can modify this algorithm for block-wise sparse models. In particular, instead of pruning based on the absolute value of each parameter and absolute value of the gradient, we use the norm 1 of each block and the norm 1 of the gradients within each block identify important blocks that needs to be updated during the training.

We keep the rank of our decomposition equal to $2$.[3] To report the standard deviation, we run our experiment five times. Column FLOPs indicate the number of FLOPs needed during forward and backward paths. We used PyTorch package `ptflops` to calculate the number of flops. This table shows that our method generally can achieve better accuracy and better sparsity rate compared to group LASSO and elastic group LASSO. The number of FLOPs and training parameters also are smaller under our algorithms. For instance, when the block size is 16 by 2, the number of FLOPs and training parameters is significantly lower than the baselines. However, our algorithm is able to achieve a higher accuracy and a higher sparsity rate compared to the baselines.

As another baseline, we compare our algorithm with unstructured pruning (Han et al., 2015) (iterative pruning in the table). Generally, unstructured pruning can achieve better accuracy compared to structured and block-wise pruning. This table shows that when block size is 2 by 2, our algorithm can achieve better accuracy and better sparsity rate compared to all the baselines including iterative pruning. For other block sizes, our algorithm decreases the number of flops and training parameters significantly compared to baselines. As a result, some of the baselines may achieve better accuracy compared to our method.

Table 1: One Linear Layer Model compression results on MNIST dataset

| Block size | Models | Accuracy | Sparsity Rate | Training Params | Training FLOPs |
|---|---|---|---|---|---|
| (2,2) | Group LASSO | $85.18 \pm 0.37$ | $49.67 \pm 0.10$ | 7.84K | 7.85k |
| (2,2) | elastic group LASSO | $80.61 \pm 0.44$ | $42.11 \pm 0.76$ | 7.84K | 7.85k |
| (2,2) | Blockwise Rigl | $86.66 \pm 0.36$ | $50.61 \pm 0.00$ | 7.84K | 7.85k |
| (2,2) | Ours | $88.97 \pm 1.50$ | $86.43 \pm 0.30$ | 5.89K | 7.84k |
| (4,2) | Group LASSO | $74.12 \pm 0.98$ | $44.38 \pm 1.09$ | 7.84K | 7.85k |
| (4,2) | elastic group LASSO | $76.66 \pm 1.59$ | $44.32 \pm 3.35$ | 7.84K | 7.85k |
| (4,2) | Blockwise Rigl | $87.13 \pm 0.44$ | $50.61 \pm 0.00$ | 7.84K | 7.85k |
| (4,2) | Ours | $81.75 \pm 0.77$ | $38.57 \pm 0.48$ | 2.96k | 3.92k |
| (8,2) | Group LASSO | $75.82 \pm 0.73$ | $40.87 \pm 0.89$ | 7.84K | 7.85k |
| (8,2) | elastic group LASSO | $80.61 \pm 0.44$ | $42.11 \pm 0.76$ | 7.84K | 7.85k |
| (8,2) | Blockwise Rigl | $87.32 \pm 0.38$ | $50.61 \pm 0.00$ | 7.84K | 7.85k |
| (8,2) | Ours | $75.08 \pm 2.05$ | $50.85 \pm 0.70$ | 1.96k | 1.50k |
| (16,2) | Group LASSO | $75.82 \pm 0.73$ | $40.87 \pm 0.89$ | 7.85k | 7.84K |
| (16,2) | elastic group LASSO | $80.61 \pm 0.44$ | $42.11 \pm 0.76$ | 7.85k | 7.84K |
| (16,2) | Blockwise Rigl | $86.95 \pm 0.35$ | $50.61 \pm 0.00$ | 7.84K | 7.85k |
| (16,2) | Ours | $81.57 \pm 2.05$ | $50.85 \pm 0.70$ | 0.80k | 0.98k |
| - | Iterative Pruning | $86.72 \pm 0.24$ | $50.06 \pm 0.46$ | 7.84K | 7.85k |

**Pattern Selection** In addition, we conduct an experiment on the effectiveness of our method in pattern selection. We use the same patterns as those used in Table 1. Our goal is to use optimization problem equation 7 to pick the pattern that achieves the highest accuracy among the patterns stated in Table 2. We set $\lambda_1 = \lambda_2 = 0.01$ and increase these parameters by 0.002 every 5 epochs. We continue training for 50 epochs. We can see in Figure 3a that the parameters associated with patterns $k = 2, 3, 4$ go to zero after 40 epochs, and only $S^{(1)}$ ($k = 1$ is associated with block size 2 by 2) remains non-zero at epoch 40. This shows that pattern $k = 1$ is the best pattern in terms of accuracy. After finding the best pattern, we can fine-tune

---

[3]We choose rank 2 to demonstrate the capability of our algorithm. Clearly, selecting a larger rank can improve the performance of our algorithm even further.

the parameters associated with this pattern to get the best accuracy. After fine-tuning for 5 epochs, the accuracy of the first pattern can achieve 88.86 %.

## 6.2 LeNet on MNIST

In this part, we conduct an experiment with the LeNet-5 Network (Lecun et al., 1998) and the MNIST dataset. Similar to the previous part, we compare our algorithm with group LASSO, elastic group LASSO, and unstructured iterative pruning. The results are shown in Table 2. Since LeNet-5 has three fully connected layers, the column *block size* indicated the block sizes for these layers. For example, $(16,8)(8,4)(4,2)$ indicates that the block size in the first layer is 16 by 8, in the second layer is 8 by 4, in the third layer is 4 by 2. The rank of the decomposition under our algorithm is 5 for all the layers in all the experiments. We can see our algorithm achieves better accuracy compared to the baselines. However, the number of FLOPs and the number of training parameters of our algorithm are significantly lower compared to baselines.

Table 2: Accuracy, sparsity rate, number of training parameters, and number of FLOPs for LeNet-5 network trained on MNIST dataset for different block sizes.

| Block-size | Methods | Accuracy | Sparsity Rate | Train Param | Train FLOPs |
|---|---|---|---|---|---|
| (16, 8) (8, 4) (4, 2) | group LASSO | 98.31 ± 0.54 | 49.43 ± 0.06 | 61k | 435.85k |
| (16, 8) (8, 4) (4, 2) | Elastic group LASSO | 98.23 ± 0.60 | 49.47 ± 1.02 | 61k | 435.85k |
| (16, 8) (8, 4) (2, 2) | Blockwise Rigl | 98.28 ± 0.01 | 50.04 ± 0.00 | 61k | 435.85k |
| (16, 8) (8, 4) (4, 2) | Ours | 98.55 ± 0.56 | 50.85 ± 0.70 | 6.2k | 270.59k |
| (8, 4) (4, 4) (2, 2) | group LASSO | 97.96 ± 0.51 | 72.97 ± 14.01 | 61k | 435.85k |
| (8, 4) (4, 4) (2, 2) | Elastic group LASSO | 98.02 ± 0.51 | 63.28 ± 12.89 | 61k | 435.85k |
| (8, 4) (4, 4) (2, 2) | Blockwise Rigl | 97.77 ± 0.09 | 50.03 ± 0.00 | 61k | 435.85k |
| (8, 4) (4, 4) (2, 2) | Ours | 99.06 ± 0.52 | 52.49 ± 1.23 | 22.6k | 287.54k |
| (4, 4) (4, 4) (2, 2) | group LASSO | 98.08 ± 0.60 | 52.58 ± 4.94 | 61k | 435.85k |
| (4, 4) (4, 4) (2, 2) | Elastic Group LASSO | 98.17 ± 0.55 | 52.77 ± 5.93 | 61K | 435.85k |
| (4, 4) (4, 4) (2, 2) | Blockwise Rigl | 97.70 ± 0.05 | 50.01 ± 0.00 | 61k | 435.85k |
| (4, 4) (4, 4) (2, 2) | Ours | 99.08 ± 0.53 | 54.02 ± 1.53 | 13.7k | 306.74k |
| (4, 4) (2, 2) (2, 2) | group LASSO | 98.08 ± 0.53 | 63.29 ± 9.21 | 61k | 435.85k |
| (4, 4) (2, 2) (2, 2) | Elastic Group LASSO | 99.08 ± 0.68 | 54.30 ± 1.59 | 61k | 435.85k |
| (4, 4) (2, 2) (2, 2) | Blockwise Rigl | 97.72 ± 0.12 | 50.01 ± 0.00 | 61k | 435.85k |
| (4, 4) (2, 2) (2, 2) | Ours | 99.08 ± 0.68 | 54.30 ± 1.59 | 9.7k | 319.34k |
| (2, 2) (2, 2) (2, 2) | group LASSO | 98.27 ± 0.73 | 49.38 ± 0.02 | 61k | 435.85k |
| (2, 2) (2, 2) (2, 2) | Elastic group LASSO | 97.58 ± 0.60 | 84.43 ± 8.43 | 61k | 435.85k |
| (2, 2) (2, 2) (2, 2) | Blockwise Rigl | 98.31 ± 0.17 | 60.03 ± 0.00 | 61k | 435.85k |
| (2, 2) (2, 2) (2, 2) | Ours | 98.66 ± 0.59 | 56.27 ± 2.71 | 6.1k | 357.74k |
| - | iterative pruning | 98.02 ± 0.82 | 58.56 ± 1.32 | 61k | 425.85k |

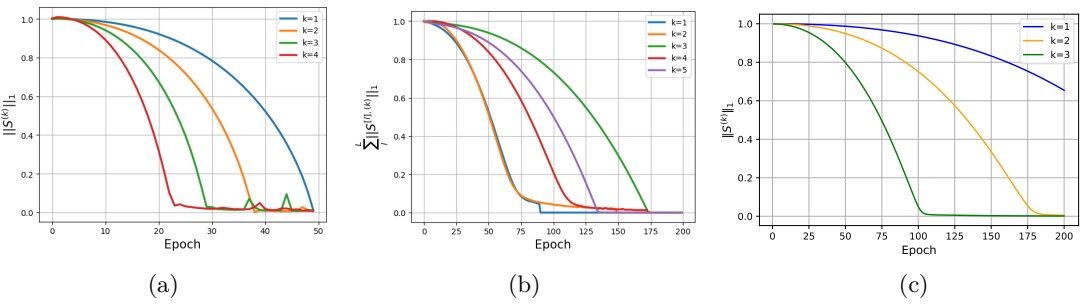

Figure 3: a) Pattern selection for a linear model. b) Pattern selection for LeNet-5 network. c) Pattern selection for ViT tiny network.

**Pattern Selection** Similarly, we also have experimented with the pattern selection. In particular, we use optimization problem in equation 7 to pick the pattern that achieves the highest accuracy among the patterns stated in Table 2. We set $\lambda_1 = \lambda_2 = 0.01$ and we increase these parameters by 0.002 every 5 epochs. We run the training for 200 epochs and we report $\sum_{l=1}^{L} ||S^{[l],(k)}||_1$ for different $1 \leq k \leq 5$ in Figure 3b. In this experiment $k = 1$ corresponds to pattern $(16, 8)(8, 4)(4, 2)$, $k = 2$ corresponds to pattern $(8, 4)(4, 4)(4, 2)$, $k = 3$ corresponds to pattern $(4, 4)(4, 4)(2, 2)$, $k = 4$ corresponds to pattern $(4, 4)(2, 2)(2, 2)$, and $k = 5$ corresponds to pattern $(2, 2)(2, 2)(2, 2)$. As we expected, the weight corresponding to the pattern $k = 3$ remains non-zero after 130 epochs. After finding the best pattern, we can fine-tune the selected pattern to achieve the maximum possible accuracy.

### 6.3 ViT/Swin-Transformer on CIFAR-100

We conduct an experiment with our approach with the ViT-tiny, ViT-base(Dosovitskiy et al., 2021; Touvron et al., 2021) and Swin-transformer Tiny (Liu et al., 2021b) on CIFAR-100 image classification dataset (He et al., 2016). The dataset has 60 thousand pictures of 100 different categories. The model is trained using this dataset for 300 epochs. We keep the rank of our algorithm equal to 4. Since most of the modules in transformer architecture are linear layers, our method can significantly decrease the number of parameters. It can be seen in Table 3 that our model's number of parameters during the training is only 3% of that of the original model for ViT-tiny. The number of training parameters also decreases by 86% for ViT-base. In the case of the Swin Transformer, we can achieve 80% compression rate during the training. We want to emphasize that both elastic group LASSO and our method can achieve high accuracy. However, as we expected, the number of training parameters and training FLOPs are significantly smaller under our proposed algorithm.

**Pattern Selection.** In this part, we validate the possibility of using equation 7 to find a pattern that achieves the highest accuracy for ViT-tiny model. We consider three patterns: pattern $k = 1$ is associated with a ViT-tiny model with block-size 2 by 2, $k = 2$ is associated with a ViT-tiny model with block-size 4 by 4, and $k = 3$ is associated with a ViT-tiny model with block-size 8 by 8. We solve optimization problem 7 and illustrates $\sum_{l=1}^{L} ||S^{[l],(k)}||_1$ in each epoch in Figure 3c. We can see the weights associated with pattern $k = 1$ is the only one that remains non-zero after 180 epochs implying that pattern $k = 1$ can achieve the highest accuracy compared to patterns $k = 2, 3$. Note that if we train ViT-tiny under these patters, we can achieve an accuracy of $64.56, 62.99, 62.88$ for $k = 1, 2, 3$, respectively. This implies that pattern $k = 1$ is the best pattern. No

Table 3: Experiment with CIFAR-100 dataset. Our method can significantly reduce the training parameters and training FLOPs while maintaining the accuracy close to the original model and (elastic) group LASSO method.

| Method | Block-size | accuracy | Sparsity Rate | Training Params | Training FLOPs |
|---|---|---|---|---|---|
| ViT-t (Original Model) | - | $64.32 \pm 1.92$ | - | 5.5M | 2.16G |
| Group LASSO | $4 \times 4$ | $60.41 \pm 4.24$ | $49.99 \pm 0.02$ | 5.5M | 2.16G |
| elastic group LASSO | $4 \times 4$ | $61.92 \pm 3.01$ | $49.92 \pm 0.11$ | 5.5M | 2.16G |
| Blockwise Rigl | $4 \times 4$ | $49.56 \pm 0.48$ | $50.67 \pm 0.00$ | 5.5M | 2.16G |
| Ours | $4 \times 4$ | $62.99 \pm 0.73$ | $49.64 \pm 0.72$ | 0.16M | 65.37M |
| ViT-b (Original Model) | - | $71.34 \pm 0.42$ | - | 87.34M | 35.34G |
| Group LASSO | $4 \times 4$ | $68.41 \pm 1.24$ | $48.92 \pm 1.34$ | 87.34M | 35.34G |
| elastic group LASSO | $4 \times 4$ | $66.95 \pm 2.17$ | $51.93 \pm 0.37$ | 87.34M | 35.34G |
| Ours | $4 \times 4$ | $69.82 \pm 0.22$ | $60.31 \pm 3.61$ | 11.58M | 10.11M |
| Swin-t(Original Model) | - | $81.44 \pm 0.05$ | - | 27.60M | 26.18G |
| Group LASSO | $4 \times 4$ | $75.87 \pm 2.17$ | $50.24 \pm 0.13$ | 27.60M | 26.18G |
| elastic group LASSO | $4 \times 4$ | $76.34 \pm 0.82$ | $50.19 \pm 0.25$ | 27.60M | 26.18G |
| Blockwise Rigl | $4 \times 4$ | $60.30 \pm 0.22$ | $50.03 \pm 0.00$ | 27.60M | 26.18G |
| Ours | $4 \times 4$ | $77.54 \pm 0.42$ | $53.25 \pm 0.36$ | 5.3M | 167.33M |

### 6.4 Ablation Experiments

Table 4: The impact of rank of the decomposition in equation 3 on the model's accuracy, sparsity rate, number of training parameters, and number of FLOPs.

| Model | Rank | accuracy | sparsity | Training Params | Training FLOPs |
|---|---|---|---|---|---|
| Linear | 1 | $48.40 \pm 0.40$ | $53.57 \pm 2.43$ | 0.26k | 0.56k |
| Linear | 2 | $66.79 \pm 0.91$ | $53.57 \pm 1.57$ | 0.46k | 1.13k |
| Linear | 4 | $84.58 \pm 3.55$ | $55.36 \pm 0.72$ | 0.85k | 2.24k |
| Linear | 6 | $88.19 \pm 0.32$ | $51.79 \pm 0.56$ | 1.24k | 3.36k |
| ViT-t | 1 | $36.86 \pm 2.41$ | $52.20 \pm 0.13$ | 0.88M | 0.54M |
| ViT-t | 2 | $59.71 \pm 2.63$ | $50.74 \pm 0.27$ | 1.22M | 1.02M |
| ViT-t | 4 | $62.99 \pm 0.73$ | $49.64 \pm 0.72$ | 1.88M | 1.88M |
| ViT-l | 1 | $53.27 \pm 1.46$ | $57.32 \pm 1.84$ | 39.19M | 34.1G |
| ViT-l | 2 | $62.79 \pm 0.43$ | $61.32 \pm 1.29$ | 58.13M | 92.51G |
| ViT-l | 4 | $73.46 \pm 0.36$ | $66.59 \pm 2.94$ | 96.01M | 329.44G |
| Swin-t | 1 | $58.46 \pm 0.16$ | $51.39 \pm 0.67$ | 3.53M | 55.78M |
| Swin-t | 2 | $68.22 \pm 0.04$ | $54.37 \pm 1.01$ | 5.25M | 108.65M |
| Swin-t | 4 | $77.54 \pm 0.42$ | $53.25 \pm 0.36$ | 8.69M | 167.33M |

In this part, we conduct an experiment to understand the impact of the rank on the accuracy, number of training parameters, and number of training flops during the forward path and backward path. We set the block size equal to $4 \times 4$. We conduct ablation experiments on the linear model, ViT-tiny, ViT-large, and Swin Transformer separately. As we expected, we can improve the accuracy of the model by increasing the rank. In this experiment, we kept the regularizer parameter the same for different ranks. As a result, the sparsity rate is not sensitive to the rank and remains almost the same for different ranks.

## 7 Conclusion

We introduce a novel approach for training block-wise sparse matrices using Kronecker product decomposition. This method offers an alternative to group LASSO/structured pruning, enabling training block-wise sparse matrices with fewer parameters and FLOPs. Our theoretical results show that our proposed method can decrease the number of training parameters and the number of FLOPs without hurting accuracy compared to the group LASSO and structured pruning algorithms. Our experiments demonstrate the effectiveness of our approach in terms of efficiency and accuracy. We further show that our algorithm enables us to automate the pattern selection and efficiently find the right block size for the sparsity pattern to achieve the best accuracy.

It is also known that the model compression can negatively impact the fairness (Tran et al., 2022). Therefore, future study is needed to study the impact of the proposed method on the fairness of the model.

## 8 Acknowledgment

This work is supported by the U.S. National Science Foundation under award IIS-2301599 and CMMI-2301601, and by grants from the Ohio State University's Translational Data Analytics Institute and College of Engineering Strategic Research Initiative.

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

# A Appendix / supplemental material

## A.1 Proof [Proposition 2]

Consider a one layer network without bias term. The input dimension of it is $n$. The output dimension of it is $m$. We can knoe the shape of the weight matrix $W$ is $\mathbb{R}^{m \times n}$. Since we have $N$ data points, the input matrix $X \in \mathbb{R}^{N \times n}$.

### A.1.1 Forward FLOPs with full matrix

*Proof.* When using full weight matrix, the first step is to compute the output $O \in \mathbb{R}^{N \times m}$ as

$$O = XW^T. \tag{8}$$

The FLOPs of this step is $Nm(2n - 1)$. Then we calculate the loss as

$$\mathcal{J} = \|O - Y\|_F^2, \tag{9}$$

where $Y \in \mathbb{R}^{N \times m}$ is the label matrix. The FLOPs for this step is $3Nm - 1$. Therefore, the FLOPs of the forward computation is

$$Nm(2n - 1) + (3Nm - 1) = \mathcal{O}\left(2Nm(n + 1)\right). \tag{10}$$

$\square$

### A.1.2 Backward FLOPs with full matrix

*Proof.* In the backward process, we need to calculate the gradient of $\mathcal{J}$ on $W$. Using chain rule, the first step is to compute

$$\frac{\partial \mathcal{J}}{\partial O} = 2(O - Y). \tag{11}$$

Since $O - Y$ has been calculated in the forward pass, the FLOPs is $Nm$. The gradient of $W$ is

$$\frac{\partial \mathcal{J}}{\partial W} = \left(\frac{\partial \mathcal{J}}{\partial O}\right)^T X. \tag{12}$$

The FLOPs for this step is $mn(2N - 1)$. Therefore, the FLOPs for the backward pass is

$$Nm + mn(2N - 1) = \mathcal{O}\left(Nm(2n + 1)\right). \tag{13}$$

$\square$

### A.1.3 Forward FLOPs with sparse matrix

With Kronecker product decomposition, we replace $W$ by $\sum_{i=1}^{r}(S \odot A_i) \otimes B_i$. $S$ and $A_i \in \mathbb{R}^{m_1 \times n_1}$, $B_i \in \mathbb{R}^{m_2 \times n_2}$, where $m_1 m_2 = m$, $n_1 n_2 = n$.

*Proof.* In the forward pass, we need to firstly reshape $X \in \mathbb{R}^{N \times n}$ into $\mathbf{reshape}(X) \in \mathbb{R}^{n_2 \times Nn_1}$. Then we calculate $B_i\mathbf{reshape}(X) \in \mathbb{R}^{m_2 \times Nn_1}$ with FLOPs $Nn_1 m_2(2n_2 - 1)$. The result is reshape into $\mathbf{reshape}(B_i\mathbf{reshape}(X)) \in \mathbb{R}^{Nm_2 \times n_1}$.

Then we calculate $S \odot A_i \in \mathbb{R}^{n_1 \times m_1}$ with FLOPs $m_1 n_1$. After this, we get

$$O_i = \mathbf{reshape}(B_i\mathbf{reshape}(X))(S \odot A_i)^T \in \mathbb{R}^{Nm_2 \times m_1}, \tag{14}$$

with FLOPs $Nm_1m_2(2n_1 - 1)$. We denote $O$ as the output of the layer, which is to say

$$O = \sum_{i=1}^{r} O_i = \sum_{i=1}^{r} \textbf{reshape}(B_i \textbf{reshape}(X))(S \odot A_i)^T. \tag{15}$$

The total FLOPs to get $O$ is

$$r(Nm_1m_2(2n_1 - 1) + m_1n_1 + Nn_1m_2(2n_2 - 1)) + (r - 1)Nm \tag{16}$$

Then we reshape $O$ in to $\textbf{reshape}(O) \in \mathbb{R}^{N \times m}$. The loss is calculated as

$$\mathcal{J} = \|\textbf{reshape}(O) - Y\|_F^2 \tag{17}$$

The FLOPs for this step is $3Nm - 1$. Therefore the FLOPs of the forward computation is

$$
\begin{aligned}
&r(2Nm_1m_2n_1 - Nm_1m_2 + m_1n_1 + 2Nm_1n_1n_2 - Nm_2n_1) + (r - 1)Nm + 3Nm - 1 \\
=&\mathcal{O}\left(2Nrm_1n_1(m_2 + n_2) - Nr(m + 2m_2n_1) + 3Nm\right)
\end{aligned} \tag{18}
$$

$\square$

### A.1.4 Backward FLOPs with sparse matrix

*Proof.* In the backward process, we need to calculate the gradient of $\mathcal{J}$ on $S, A_i$ and $B_i$. Using chain rule, the first step is to compute

$$\frac{\partial \mathcal{J}}{\partial \textbf{reshape}(O)} = 2(\textbf{reshape}(O) - Y). \tag{19}$$

Since $\textbf{reshape}(O) - Y$ has been calculated in the forward pass, the FLOPs is $Nm$. Then we reshape it into $\frac{\partial \mathcal{J}}{\partial O} \in \mathbb{R}^{Nm_2 \times m_1}$. Then we can get

$$\frac{\partial \mathcal{J}}{\partial(S \odot A_i)} = \left(\frac{\partial \mathcal{J}}{\partial O}\right)^T \textbf{reshape}(B_i \textbf{reshape}(X)). \tag{20}$$

Since $\textbf{reshape}(B_i \textbf{reshape}(X))$ has been obtained in the forward pass, the FLOPs for this step is $m_1n_1(2Nm_2 - 1)$. To get the gradient on $S$ and $A_i$, we have

$$\frac{\partial \mathcal{J}}{\partial S} = \sum_{i=1}^{r} \frac{\partial \mathcal{J}}{\partial(S \odot A_i)} \odot A_i, \tag{21}$$

with FLOPs $rm_1n_1 + (r - 1)m_1n_1$, and

$$\frac{\partial \mathcal{J}}{\partial A_i} = \frac{\partial \mathcal{J}}{\partial(S \odot A_i)} \odot S, \tag{22}$$

with FLOPs $m_1n_1$. The gradient on $\textbf{reshape}(B_i \textbf{reshape}(X))$ is

$$\frac{\partial \mathcal{J}}{\partial \textbf{reshape}(B_i \textbf{reshape}(X))} = \frac{\partial \mathcal{J}}{\partial O}(S \odot A_i). \tag{23}$$

The FLOPs for this step is $Nm_2n_1(2m_1 - 1)$. We reshape the gradient into $\frac{\partial \mathcal{J}}{\partial B_i \textbf{reshape}(X)} \in \mathbb{R}^{m_2 \times Nn_1}$. So, we can get the gradient on $B_i$ as

$$\frac{\partial \mathcal{J}}{\partial B_i} = \frac{\partial \mathcal{J}}{\partial B_i \textbf{reshape}(X)} \textbf{reshape}(X)^T, \tag{24}$$

with FLOPs of $m_2n_2(2Nn_1 - 1)$. Therefore, we can get the total FLOPs for the backward pass as

$$
\begin{aligned}
&Nm + rm_1n_1(2Nm_2 - 1) + rm_1n_1 + (r - 1)m_1n_1 + rm_1n_1 + rNm_2n_1(2m_1 - 1) + rm_2n_2(2Nn_1 - 1) \\
=&\mathcal{O}\left(Nm + Nr(4m_1m_2n_1 - m_2n_1 + 2m_2n_1n_2)\right)
\end{aligned} \tag{25}
$$

$\square$

### A.2 Proof [Proposition 3]

Consider a two-layer network without bias term. The input dimension of the first linear layer is $m^{[1]}$. The output dimension of the linear layer is $m^{[2]}$. So, the input dimension for the second linear layer is $m^{[2]}$. The output dimension of the linear layer is $m^{[3]}$. With the network architecture, we can know that $W^{[1]} \in \mathbb{R}^{m^{[2]} \times m^{[1]}}$ and $W^{[2]} \in \mathbb{R}^{m^{[3]} \times m^{[2]}}$. Since we have $N$ data points, for the input matrix of the first layer denoted by $X^{[1]}$, we have, $X^{[1]} \in \mathbb{R}^{N \times m^{[1]}}$. We use $\sigma$ to represent the activation function for the first layer.

#### A.2.1 Forward FLOPs with full matrix

*Proof.* When using full weight matrix, the first step is to compute the pre-activated output $O^{[1]} \in \mathbb{R}^{N \times m^{[2]}}$ as

$$O^{[1]} = X^{[1]} W^{[1]T}. \tag{26}$$

The FLOPs of this step is $Nm^{[2]}(2m^{[1]} - 1)$. Then, we compute the activation $X^{[2]} \in \mathbb{R}^{N \times m^{[2]}}$ as follows,

$$X^{[2]} = \sigma(O^{[1]}). \tag{27}$$

The FLOPs for calculating the activation is $Nm^{[2]}$. For the second layer, we need to compute $O^{[2]} \in \mathbb{R}^{N \times m^{[3]}}$ as $O^{[2]} = X^{[2]} W^{[2]T}$. The FLOPs are $Nm^{[3]}(2m^{[2]} - 1)$. The last step is calculating the loss as

$$\mathcal{J} = \left\| O^{[2]} - Y \right\|_F^2, \tag{28}$$

where $Y \in \mathbb{R}^{N \times m^{[3]}}$ is the label matrix. The FLOPs for this step is $3Nm^{[3]} - 1$. Therefore, the FLOPs of the forward computation is

$$Nm^{[2]}(2m^{[1]} - 1) + Nm^{[2]} + Nm^{[3]}(2m^{[2]} - 1) + 3Nm^{[3]} - 1 = 2Nm^{[1]}m^{[2]} + 2Nm^{[2]}m^{[3]} + 2Nm^{[3]} - 1$$
$$= \mathcal{O}\left( 2N(m^{[1]}m^{[2]} + m^{[2]}m^{[3]} + m^{[3]}) \right). \tag{29}$$

$\square$

#### A.2.2 Backward FLOPs with full matrix

*Proof.* In the backward process, we need to calculate the gradient of $\mathcal{J}$ on $W^{[1]}$ and $W^{[2]}$. Using chain rule, the first step is to compute

$$\frac{\partial \mathcal{J}}{\partial O^{[2]}} = 2(O^{[2]} - Y). \tag{30}$$

Since $O^{[2]} - Y$ has been calculated in the forward pass, the FLOPs is $Nm^{[3]}$. The gradient of $W^{[2]}$ is

$$\frac{\partial \mathcal{J}}{\partial W^{[2]}} = \left( \frac{\partial \mathcal{J}}{\partial O^{[2]}} \right)^T X^{[2]}. \tag{31}$$

The FLOPs for this step is $m^{[2]}m^{[3]}(2N - 1)$. To compute the gradient on $W^{[1]}$, we need to first compute

$$\frac{\partial \mathcal{J}}{\partial X^{[2]}} = \left( \frac{\partial \mathcal{J}}{\partial O^{[2]}} \right) W^{[2]}, \tag{32}$$

with the FLOPs $Nm^{[2]}(2m^{[3]} - 1)$. Then compute

$$\frac{\partial \mathcal{J}}{\partial O^{[1]}} = \left( \frac{\partial \mathcal{J}}{\partial X^{[2]}} \right) \odot (\sigma(O^{[1]})), \tag{33}$$

with the FLOPs $Nm^{[2]}$. The final step is

$$\frac{\partial \mathcal{J}}{\partial W^{[1]}} = \left(\frac{\partial \mathcal{J}}{\partial O^{[1]}}\right)^T X^{[1]} \tag{34}$$

The FLOPs is $m^{[1]}m^{[2]}(2N-1)$. Therefore, the FLOPs for the backward pass is

$$Nm^{[3]} + m^{[2]}m^{[3]}(2N-1) + Nm^{[2]}(2m^{[3]} - 1) + Nm^{[2]} + m^{[1]}m^{[2]}(2N-1)$$
$$= 2Nm^{[1]}m^{[2]} + 4Nm^{[2]}m^{[3]} + Nm^{[3]} - m^{[1]}m^{[2]} - m^{[2]}m^{[3]}$$
$$= \mathcal{O}\left(N(2m^{[1]}m^{[2]} + 4m^{[2]}m^{[3]} + m^{[3]})\right). \tag{35}$$

$\square$

### A.2.3 Forward FLOPs with sparse matrix

With the Kronecker product decomposition, we replace $W^{[1]}$ by $\sum_{i=1}^{r^{[1]}}(S^{[1]} \odot A_i^{[1]}) \otimes B_i^{[1]}$ and $W^{[2]}$ by $\sum_{i=1}^{r^{[2]}}(S^{[2]} \odot A_i^{[2]}) \otimes B_i^{[2]}$. In the first layer, $S^{[1]}$ and $A_i^{[1]} \in \mathbb{R}^{m_1^{[2]} \times m_1^{[1]}}$, $B_i^{[1]} \in \mathbb{R}^{m_2^{[2]} \times m_2^{[1]}}$, where $m_1^{[1]}m_2^{[1]} = m^{[1]}$, $m_1^{[2]}m_2^{[2]} = m^{[2]}$. In the second layer, $S^{[2]}$ and $A_i^{[2]} \in \mathbb{R}^{m_1^{[3]} \times m_1^{[2]}}$, $B_i^{[2]} \in \mathbb{R}^{m_2^{[3]} \times m_2^{[2]}}$, where $m_1^{[3]}m_2^{[3]} = m^{[3]}$.

*Proof.* In the forward pass, we need to firstly reshape $X^{[1]} \in \mathbb{R}^{N \times m^{[1]}}$ into $\textbf{reshape}(X^{[1]}) \in \mathbb{R}^{m_2^{[1]} \times Nm_1^{[1]}}$. Then we calculate $B_i^{[1]}\textbf{reshape}(X^{[1]}) \in \mathbb{R}^{m_2^{[2]} \times Nm_1^{[1]}}$ with the FLOPs $Nm_1^{[1]}m_2^{[2]}(2m_2^{[1]} - 1)$. The result is reshaped into $\textbf{reshape}(B_i^{[1]}\textbf{reshape}(X^{[1]})) \in \mathbb{R}^{Nm_2^{[2]} \times m_1^{[1]}}$.

Then we calculate $S^{[1]} \odot A_i^{[1]} \in \mathbb{R}^{m_1^{[2]} \times m_1^{[1]}}$ with the FLOPs $m_1^{[1]}m_1^{[2]}$. After this, we get

$$O_i^{[1]} = \textbf{reshape}(B_i^{[1]}\textbf{reshape}(X^{[1]}))(S^{[1]} \odot A_i^{[1]})^T \in \mathbb{R}^{Nm_2^{[2]} \times m_1^{[2]}}, \tag{36}$$

with FLOPs $Nm_2^{[2]}m_1^{[2]}(2m_1^{[1]} - 1)$. We denote $O^{[1]}$ as the pre-activated result of the first layer, which is to say

$$O^{[1]} = \sum_{i=1}^{r^{[1]}} O_i^{[1]} = \sum_{i=1}^{r^{[1]}} \textbf{reshape}(B_i^{[1]}\textbf{reshape}(X^{[1]}))(S^{[1]} \odot A_i^{[1]})^T. \tag{37}$$

The total FLOPs to get $O^{[1]}$ is

$$r^{[1]}\left(Nm_1^{[1]}m_2^{[2]}(2m_2^{[1]} - 1) + m_1^{[1]}m_1^{[2]} + Nm_2^{[2]}m_1^{[2]}(2m_1^{[1]} - 1)\right) + (r^{[1]} - 1)m^{[2]}N$$
$$= r^{[1]}\left(2Nm_1^{[1]}m_2^{[2]} + 2Nm^{[2]}m_1^{[1]} - Nm_2^{[2]}(m_1^{[1]} + m_1^{[2]}) + m_1^{[1]}m_1^{[2]}\right) + (r^{[1]} - 1)m^{[2]}N. \tag{38}$$

The input for the second layer $X^{[2]} \in \mathbb{R}^{Nm_2^{[2]} \times m_1^{[2]}}$ is obtained by

$$X^{[2]} = \sigma(O^{[1]}), \tag{39}$$

with FLOPs $Nm_2^{[2]}m_1^{[2]} = Nm^{[2]}$. Then we reshape it into $\textbf{reshape}(X^{[2]}) \in \mathbb{R}^{m_2^{[2]} \times Nm_1^{[2]}}$. We calculate $B_i^{[2]}\textbf{reshape}(X^{[2]}) \in \mathbb{R}^{m_2^{[3]} \times Nm_1^{[2]}}$ with the FLOPs $Nm_1^{[2]}m_2^{[3]}(2m_2^{[2]} - 1)$. The result is reshaped into $\textbf{reshape}(B_i^{[2]}\textbf{reshape}(X^{[2]}) \in \mathbb{R}^{Nm_2^{[3]} \times m_1^{[2]}}$.

Similar to the first layer, we calculate $S^{[2]} \odot A_i^{[2]} \in \mathbb{R}^{m_1^{[3]} \times m_1^{[2]}}$ with the FLOPs $m_1^{[2]}m_1^{[3]}$. After this, we get

$$O_i^{[2]} = \textbf{reshape}(B_i^{[2]}\textbf{reshape}(X^{[2]}))(S^{[2]} \odot A_i^{[2]})^T \in \mathbb{R}^{Nm_2^{[3]} \times m_1^{[3]}}, \tag{40}$$

with FLOPs $Nm_2^{[3]}m_1^{[3]}(2m_1^{[2]}-1)$. We denote $O^{[2]}$ as the output of the second layer, which is to say

$$O^{[2]} = \sum_{i=1}^{r^{[2]}} O_i^{[2]} = \sum_{i=1}^{r^{[2]}} \mathbf{reshape}(B_i^{[2]}\mathbf{reshape}(X^{[2]}))(S^{[2]} \odot A_i^{[2]})^T. \tag{41}$$

The total FLOPs to get $O^{[2]}$ is

$$r^{[2]}\left(Nm_1^{[2]}m_2^{[3]}(2m_2^{[2]}-1) + m_1^{[2]}m_1^{[3]} + Nm_2^{[3]}m_1^{[3]}(2m_1^{[2]}-1)\right) + (r^{[2]}-1)m^{[3]}N$$

$$=r^{[2]}\left(2Nm^{[2]}m_2^{[3]} + 2Nm^{[3]}m_1^{[2]} - Nm_2^{[3]}(m_1^{[2]}+m_1^{[3]}) + m_1^{[2]}m_1^{[3]}\right) + (r^{[2]}-1)m^{[3]}N. \tag{42}$$

Then we shape $O^{[2]}$ into $\mathbf{reshape}(O^{[2]}) \in \mathbb{R}^{N \times m^{[3]}}$. The loss is calculated as

$$\mathcal{J} = \left\|\mathbf{reshape}(O^{[2]}) - Y\right\|_F^2. \tag{43}$$

The FLOPs for this step is $3Nm^{[3]} - 1$. Therefore, the FOLPs of the forward computation is

$$r^{[1]}\left(2Nm^{[1]}m_2^{[2]} + 2Nm^{[2]}m_1^{[1]} - Nm_2^{[2]}(m_1^{[1]}+m_1^{[2]}) + m_1^{[1]}m_1^{[2]}\right) + (r^{[1]}-1)m^{[2]}N + Nm^{[2]}+$$

$$r^{[2]}\left(2Nm^{[2]}m_2^{[3]} + 2Nm^{[3]}m_1^{[2]} - Nm_2^{[3]}(m_1^{[2]}+m_1^{[3]}) + m_1^{[2]}m_1^{[3]}\right) + (r^{[2]}-1)m^{[3]}N + 3Nm^{[3]} - 1$$

$$=\mathcal{O}\left(r^{[1]}\left(2Nm^{[1]}m_2^{[2]} + 2Nm^{[2]}m_1^{[1]} - Nm_2^{[2]}(m_1^{[1]}+m_1^{[2]})\right) + r^{[2]}\left(2Nm^{[2]}m_2^{[3]} + 2Nm^{[3]}m_1^{[2]} - Nm_2^{[3]}(m_1^{[2]}+m_1^{[3]})\right) + Nm^{[2]} + 3Nm^{[3]}\right) \tag{44}$$

Let $C_1 = 2Nm^{[1]}m_2^{[2]} + 2Nm^{[2]}m_1^{[1]} - Nm_2^{[2]}(m_1^{[1]}+m_1^{[2]})$, $C_2 = 2Nm^{[2]}m_2^{[3]} + 2Nm^{[3]}m_1^{[2]} - Nm_2^{[3]}(m_1^{[2]}+m_1^{[3]})$, we have the FLOPs as $\mathcal{O}\left(r^{[1]}C_1 + r^{[2]}C_2 + Nm^{[2]} + 3Nm^{[3]}\right)$. $\quad\square$

### A.2.4 Backward FLOPs with sparse matrix

*Proof.* In the backward process, we need to calculate the gradient of $\mathcal{J}$ on $S^{[1]}, A_i^{[1]}, B_i^{[1]}, S^{[2]}, A_i^{[2]}$ and $B_i^{[2]}$. Using chain rule, the first step is to compute

$$\frac{\partial \mathcal{J}}{\partial\, \mathbf{reshape}(O)^{[2]}} = 2(\mathbf{reshape}(O^{[2]}) - Y). \tag{45}$$

Since $\mathbf{reshape}(O^{[2]}) - Y$ has been calculated in the forward pass, the FLOPs is $Nm^{[3]}$. Then we reshape it into $\frac{\partial \mathcal{J}}{\partial O^{[2]}} \in \mathbb{R}^{Nm_2^{[3]} \times m_1^{[3]}}$. Then we can get

$$\frac{\partial \mathcal{J}}{\partial(S^{[2]} \odot A_i^{[2]})} = (\frac{\partial \mathcal{J}}{\partial O^{[2]}})^T\mathbf{reshape}(B_i^{[2]}\mathbf{reshape}(X^{[2]})). \tag{46}$$

Since $\mathbf{reshape}(B_i^{[2]}\mathbf{reshape}(X^{[2]}))$ has been obtained in the forward pass, the FLOPs for this step is $m_1^{[2]}m_1^{[3]}(2Nm_2^{[3]}-1)$. To get the gradient on $S^{[2]}$ and $A_i^{[2]}$, we have

$$\frac{\partial \mathcal{J}}{\partial S^{[2]}} = \sum_{i=1}^{r^{[2]}} \frac{\partial \mathcal{J}}{\partial(S^{[2]} \odot A_i^{[2]})} \odot A_i^{[2]}, \tag{47}$$

with FLOPs $r^{[2]}m_1^{[2]}m_1^{[3]} + r^{[2]} - 1$, and

$$\frac{\partial \mathcal{J}}{\partial A^{[2]}} = \frac{\partial \mathcal{J}}{\partial(S^{[2]} \odot A_i^{[2]})} \odot S^{[2]}, \tag{48}$$

with FLOPs $m_1^{[2]} m_1^{[3]}$. The gradient on $\mathbf{reshape}(B_i^{[2]}\mathbf{reshape}(X^{[2]}))$ is

$$\frac{\partial \mathcal{J}}{\partial \ \mathbf{reshape}(B_i^{[2]}\mathbf{reshape}(X^{[2]}))} = \frac{\partial \mathcal{J}}{\partial O^{[2]}}(S^{[2]} \odot A_i^{[2]}). \tag{49}$$

The FLOPs for this step is $N m_1^{[2]} m_2^{[3]}(2m_1^{[3]} - 1)$. We reshape the gradient into $\frac{\partial \mathcal{J}}{\partial B_i^{[2]}\mathbf{reshape}(X^{[2]})} \in$ $\mathbb{R}^{m_2^{[3]} \times N m_1^{[2]}}$. So, we can get the gradient on $B_i^{[2]}$ as

$$\frac{\partial \mathcal{J}}{\partial B_i^{[2]}} = \frac{\partial \mathcal{J}}{\partial B_i^{[2]}\mathbf{reshape}(X^{[2]})}\mathbf{reshape}(X^{[2]})^T, \tag{50}$$

with the FLOPs of $m_2^{[2]} m_2^{[3]}(2N m_1^{[2]} - 1)$. The gradient on $\mathbf{reshape}(X^{[2]})$ is

$$\frac{\partial \mathcal{J}}{\partial \ \mathbf{reshape}(X^{[2]})} = \sum_{i=1}^{r^{[2]}} B_i^{[2]T}\frac{\partial \mathcal{J}}{\partial B_i^{[2]}\mathbf{reshape}(X^{[2]})}, \tag{51}$$

with the FLOPs $r^{[2]}(m_1^{[2]} N m_2^{[2]}(2m_2^{[3]} - 1) + (r^{[2]} - 1)N m^{[2]}$. Then we reshape it to get $\frac{\partial \mathcal{J}}{\partial X^{[2]}} \in \mathbb{R}^{N m_2^{[2]} \times m_1^{[2]}}$. Then we calculate the gradient on $O^{[1]}$ as

$$\frac{\partial \mathcal{J}}{\partial O^{[1]}} = \left(\frac{\partial \mathcal{J}}{\partial X^{[2]}}\right) \odot (\sigma(O^{[1]})), \tag{52}$$

with FLOPs $N m^{[2]}$. Then we can get

$$\frac{\partial \mathcal{J}}{\partial (S^{[1]} \odot A_i^{[1]})} = (\frac{\partial \mathcal{J}}{\partial O^{[1]}})^T \mathbf{reshape}(B_i^{[1]}\mathbf{reshape}(X^{[1]})). \tag{53}$$

The FLOPs for this step is $m_1^{[1]} m_1^{[2]}(2N m_2^{[2]} - 1)$. So, the gradients on $S^{[1]}$ and $A_i^{[1]}$ are

$$\frac{\partial \mathcal{J}}{\partial S^{[1]}} = \sum_{i=1}^{r^{[1]}} \frac{\partial \mathcal{J}}{\partial (S^{[1]} \odot A_i^{[1]})} \odot A_i^{[1]}, \tag{54}$$

with FLOPs $r^{[1]} m_1^{[1]} m_1^{[2]} + (r^{[1]} - 1)m_1^{[1]} m_1^{[2]}$, and

$$\frac{\partial \mathcal{J}}{\partial S^{[1]}} = \frac{\partial \mathcal{J}}{\partial (S^{[1]} \odot A_i^{[1]})} \odot S^{[1]}, \tag{55}$$

with FLOPs $m_1^{[1]} m_1^{[2]}$. The gradient on $\mathbf{reshape}(B_i^{[1]}\mathbf{reshape}(X^{[1]})$ is

$$\frac{\partial \mathcal{J}}{\partial \ \mathbf{reshape}(B_i^{[1]}\mathbf{reshape}(X^{[1]}))} = \frac{\partial \mathcal{J}}{\partial O^{[1]}}(S^{[1]} \odot A_i^{[1]}). \tag{56}$$

The FLOPs for this step is $N m_1^{[1]} m_2^{[2]}(2m_1^{[2]} - 1)$. Then we reshape it to get $\frac{\partial \mathcal{J}}{\partial B_i^{[1]}\mathbf{reshape}(X^{[1]})} \in \mathbb{R}^{m_2^{[2]} \times N m_1^{[1]}}$. So, we can get the gradient on $B_i^{[1]}$ as

$$\frac{\partial \mathcal{J}}{\partial B_i^{[1]}} = \frac{\partial \mathcal{J}}{\partial B_i^{[1]}\mathbf{reshape}(X^{[1]})}\mathbf{reshape}(X^{[1]})^T, \tag{57}$$

with FLOPs $m_2^{[1]} m_2^{[2]}(2N m_1^{[1]} - 1)$.

Therefore, we can get the total FLOPs for the backward pass as

$$
\begin{aligned}
&Nm^{[3]} + r^{[2]}(m_1^{[2]}m_1^{[3]})(2Nm_2^{[3]} - 1) + r^{[2]}m_1^{[2]}m_1^{[3]} + (r^{[2]} - 1)Nm^{[2]} + r^{[2]}m_1^{[2]}m_1^{[3]} + r^{[2]}Nm_1^{[2]}m_2^{[3]}(2m_1^{[3]} - 1) + \\
&r^{[2]}m_2^{[2]}m_2^{[3]}(2Nm_1^{[2]} - 1) + r^{[2]}(m_1^{[2]}m_2^{[2]}(2m_2^{[3]} - 1) + (r^{[2]} - 1)Nm^{[2]} + Nm^{[2]} + r^{[1]}(Nm_1^{[1]}m_1^{[2]}(2Nm_2^{[2]}) - 1) + \\
&r^{[1]}m_1^{[1]}m_1^{[2]} + (r^{[1]} - 1)m_1^{[1]}m_1^{[2]} + r_1 m_1^{[1]}m_1^{[2]} + r^{[1]}Nm_1^{[1]}m_2^{[2]}(2m_1^{[2]} - 1) + r^{[1]}m_2^{[1]}m_2^{[2]}(2Nm_1^{[1]} - 1) \\
=&\mathcal{O}\left(N(m^{[2]} + m^{[3]}) + r^{[2]}Nm_1^{[2]}(4m^{[3]} - m_2^{[3]}) + 2r^{[2]}Nm^{[2]}m_2^{[3]} + r^{[1]}Nm_1^{[3]}(4m^{[2]} - m_2^{[2]}) + 2r^{[1]}Nm^{[1]}m_2^{[2]}\right).
\end{aligned}
$$

$$(58)$$

$\square$

## B    Computation resource

we used a server with 64 CPUs of AMD EPYC 7313 16-Core Processor. The server has 8 RTX A5000 GPUs, with 24GB memory for each one. For the experiment with linear model and LeNet, we used only one single GPU. And for the ViT-tiny experiment, we use 2 GPUs at the same time.

