# OpenReview forum: "An Efficient Training Algorithm for Models with Block-wise Sparsity"
_TMLR — Accepted by TMLR_

### Review · Reviewer_dSGQ · 2024-12-16

**Summary Of Contributions:**

This work introduces an efficient training algorithm with block-wise sparse weight matrices. Key contributions include:

  - **Matrix decomposition**: The study proposes to use Kronecker Product Decomposition (KPD) to decompose weight matrices into block-wise sparse matrices for efficient training.  It represents weight matrices with fewer parameters and reduces FLOPs.
  - **Theoretical analysis**: The study provides theoretical grounding that any block-wise matrices can be decomposed using KPD.
  - **Block size selection**: The study proposes an algorithm that efficiently find the optimal block size for the block-wise sparse matrices with only one round of training.
  - **Experiments**: The study validates its work with training experiments on MNIST and CIFAR-100 datasets using linear models, LeNet-5, ViT, and Swin and claims to achieve better or similar accuracy with lower FLOPs and less training parameters.

**Audience:**

Yes

**Claims And Evidence:**

Yes

**Requested Changes:**

1. Metrics:
    - To claim that the proposed algorithm can reduce training time, it needs to show the training time, not just FLOPs or number of parameters. I suggest to include some training time numbers in the experiment.
1. More experiments
    - As discussed in Weaknesses section, I'm not convinced that this would work with larger models. (1) How does rank affect with larger model? Do we need large rank not to lose accuracy? (2) Does loss function with pattern selection would work with larger model? How do you find the optimal pattern? (3) How does added regularization terms affect the step time and convergence for larger models? Maybe you can re-use ViT and Swin experiments, but further develop it.
1. Address limitations
    - Add a section discussing limitations of the proposed method.
1. Clarity (Nit)
    - Needs consistent usage between lasso and LASSO
    - 60 thousand -> 60,000 (p10)
    - Change k=2 corresponds to pattern `(8,4)(4,4)(4,2)` to `(8,4)(4,4)(2,2)` (p10)
    - **Table 4**: When discussing the impact of rank of the decomposition, why LeNet-5 is not included in this ablation experiment? Is there a specific reason?


I really like the idea of selecting optimal block size with just one training round. If you can have some back-ups to show that it works with medium sized models that may have 10s or 100s possible sparsity patterns, that would be great.

**Strengths And Weaknesses:**

### Strengths
- **Theoretical grounding**: The idea of using KPD for training of block-wise sparse models is straightforward and great. This study also provides a solid theoretical grounding on how block-wise sparse matrices can be represented using KPD, and what is theoretical FLOPs savings that it can achieve.

- **block size selection**: It is a novel idea to find optimal block size by having multiple patterns as a regularization term. It can significantly reduce searching time or make block-size searching more systematic.

### Weaknesses:
- **Practical limitation** : Although this work has thorough theoretical grounding in regards to the reduction in training parameters, it has some practical limitations. Since LeNet-5 has 3 FC layers, 5 different patterns might be enough. However, the number of different patterns increase with the number of layers. I doubt the loss function with pattern selection would work with large K value in regularization term. If #patterns K is not scalable and has to be limited (e.g. K <= 5), then the proposed algorithm cannot find the optimal block size in one round.

- **Inappropriate metric** : This study uses `FLOPs` and training parameters (`Train Param`) as metrics to indicate the reduction in training time. Lower `FLOPs` does not mean anything if it does not lead to reduced computation time. For instance Nvidia H100 supports 2:4 sparsity, where two zero weight values in contiguous block of four weight values can be efficiently accelerated. The study does not show how different block sizes can actually reduce the training time (or the step time).

- **Limited scope of experiments**:
  - The experiments uses MNIST and CIFAR-100 dataset and very outdated small neural networks such as Linear model and LeNet-5. More evaluations on large-scale datasets are needed. In particular, modifying regularization (such as this work) has a high impact on the number of training steps needed. Worse, if the training dataset is scarce, increasing the effect of regularization may lead the training to not converge at all. As pattern selection logic is part of regularization term in the equation, thorough evaluations on larger dataset and model is definitely needed to see if the new loss function with pattern selection is robust.
  - Also for larger models and datasets, I wonder how the accuracy would drop with the proposed algorithm.
  - VitT and Swin experiments are weak. How did you choose block-size for this experiment? Did you use pattern selection optimization or just choose one block size? You can always choose the block-size that works best for your work, but not for others, so more explanation is needed.

---

> ### Author Response · Authors · 2024-12-29
>
> **Practical limitation**: In the footnote of page 7, we mentioned that it is common to use the same block size across different layers and choose a block size that is an exponent of 2 [1,2,3]. So, it is very likely that K is around 100. Take Llama 3 as an example, the hidden dimension is $4096=2^{12}$. If we only consider square blocks, except for block size = (1,1) and (4096, 4096), there are only ten possible choices for the sparsity pattern. If we consider rectangular blocks, then we have K = 142. For large K, our proposed algorithm (optimization (7)) enables us to use **divide and conquer** to accelerate pattern selection procedure. For example, for K=142, we only need to do 13 rounds of pattern selection (We divide the patterns  into 12 groups, and run optimization problem (7) for each group and get the best pattern in each group. Then we can run optimization problem (7) to find the best block size among the 12 groups).
>
> **Inappropriate metric**: Training time does not always stand for efficiency. For example, consider a one layer neural network with $10^6$ parameters and a three layer neural network with 300 parameters (100 parameters for each layer). Using our GPU, we can see that the forward path of one layer neural network  takes 4 millisecond while the forward path of three layer neural network takes 14 millisecond. However, the three layer neural network has much fewer parameters and is lighter and needs less resources. In a three-layer neural network, the operations in the second layer can be done after the operations of the first layer. This is the reason that the forward path takes longer in a small three-layer neural network compared to a large one layer neural network.
>
> The above example is applicable to Kronecker product decomposition as well. Using Kronecker product decomposition, We replace $Wx$ by $B \check{x} A^T$ (check the paragraph after equation 2). Operation $B \check{x}  A^T$ is a lighter operation. However, $B \check{x}  A^T$ operation introduces two smaller linear layers instead of one larger linear layer of $Wx$. As a result, while  $B \check{x}  A^T$  is a much lighter operation, it may take longer.
>
> In this paper, we did not claim that we want to decrease the training time. Instead, we want to decrease the number of FLOPs and number of parameters during training. So we can run training in smaller or less expensive devices (e.g., edge devices).
>
> Note that, the kronecker product decomposition in equation (3) is used only for training. Using the decomposition in equation (3), If $S^{[l]}$ is a sparse matrix, then $W^{[l]}$ will be a blockwise sparse matrix. Therefore, during training, we find $S^{[l]}, A_i^{[l]}, B_i^{[l]}$ (this decreases the number of training parameters significantly). Then, during inference, we can use blockwise sparse matrix $W^{[l]} = \sum_{i=1}^{r_l} (S^{[l]}\odot A^{[l]}_i)\otimes B^{[l]}_i$ directly.
>
>
>
> **Limited scope of experiments**:
> we conduct our experiment on the tiny-imagenet
> We repeat the experiment of table 3 on the ViT-base model with 86M and ViT-Large with 307M parameters. here are the results:
>
> – Our method for ViT-base introduces 27.65M training parameters and achieves 69.82% accuracy and 57.80% sparsity (Block size is 4x4).
>
> – Our method on ViT-large introduces 59.79M training parameters and achieves 69.91% accuracy and 78.18% sparsity rate (Block size is 4x4).
>
> We chose the block size to be 4x4 in ViT and Swin experiment. This is because the block size generally should be an exponent of 2 [1,2,3]. We wanted to show that under the same block size, our method would be more efficient compated to baselines. Please let us know if you want us to extend table 3 to other block-sizes.
>
>
>
> We are running the experiments of table 4 and experiment of figure 3 on ViT-base with 86M parameters. Once we are done with those experiments, we will submit another comment.
>
>
>
>
> **Request Changes**:
>
> **Address Limitation**:
> Based on our answer to weakness 1, our method decreases the number of training FLOPs and training parameters but does not decrease the training time. We will add this in the limitation section.
>
>
> **Clarity**:
> Thanks for your comment, We will revise the error in the following edition.
>
> **Table 4**: We will add ablation study for Lenet-5 as well.
>
> [1]Peng Xu, Wenqi Shao, Mengzhao Chen, Shitao Tang, Kaipeng Zhang, Peng Gao, Fengwei An, Yu Qiao, and Ping Luo. Besa: Pruning large language models with blockwise parameter-efficient sparsity allocation, 2024. URL https://arxiv.org/abs/2402.16880.
>
> [2]Scott Gray, Alec Radford, and Diederik P. Kingma. Gpu kernels for block-sparse weights. 2017. URL https://api.semanticscholar.org/CorpusID:52220661.
>
> [3] Paolo D’Alberto, Taehee Jeong, Akshai Jain, Shreyas Manjunath, Mrinal Sarmah, Samuel Hsu, Yaswanth Raparti, and Nitesh Pipralia. Weight block sparsity: Training, compilation, and ai engine accelerators, 2024. URL https://arxiv.org/abs/2407.09453.

---

> > ### Comment · Reviewer_dSGQ · 2025-01-06
> >
> > Thanks for the reply.
> > Also, thanks for adding changes and addressing limitations.
> >
> > 1. For large K, divide and conquer approach makes sense. Thanks for the explanation.
> > 2. I understand that reduced #parameter enables smaller devices to fit the model and run training. In that regards, using #param as one of metrics makes sense. But I'm still not convinced #FLOPs itself is a good metric. As you mentioned in your reply, reduced #FLOPs can lead to increased training time, but the paper claims that it achieves reduced #FLOPs. Readers might expect reduced training time with reduced #FLOPs, but it depends on the utilization factor. If the blockwise sparsity is not optimal for GPU to compute, it leads to lower utilization and can even result in increased training time. It would be great to see how compute utilization changes with blockwise sparsity (a single layer would be enough). In that way, readers can at least expect the reduction in training time.
> > 3. I appreciate running more experiments for table 3 and table 4.
> >
> > Thanks.

---

> > > ### Author Response · Authors · 2025-01-08
> > >
> > > Thanks for your reply. We want to clarify that we use decomposed matrices $A^{[l]}_i$ and $B^{[l]}_i$ in the training time, and then we use the blockwise matrix $W^{[l]} $ only in the inference time. The number of #FLOPs is reduced during training because of decomposition. During the inference time, we use blockwise sparse matrices to decrease the computation. There are some researches showing acceleration can be achieved by using the blockwise sparse matrices during inference [1,2,3,4].
> > >
> > > Since our method and baseline are using a block-wise sparse model during inference, the speed up during inference by our method should be the same as that by the baselines. However, our method improves efficiency during training through decomposition.
> > >
> > > For completeness, to get an idea how much speed up we can get during inference, we measured the time required for a matrix multiplication using this [library](https://github.com/huggingface/pytorch_block_sparse). We adopted a sparsity rate of 0.75 and block size of 32, and a 4096 by 4096 matrix and tested 1000 times each. A Matrix-vector multiplication takes about $0.156 \pm 0.098$ sec for a dense weight matrix and takes $0.074 \pm 0.061$ sec for a blockwise sparse weight matrix.
> > >
> > >
> > > [1]Trevor Gale, Matei Zaharia, Cliff Young, and Erich Elsen. Sparse gpu kernels for deep learning,2020. URL https://arxiv.org/abs/2006.10901.
> > >
> > > [2]Paolo D’Alberto, Taehee Jeong, Akshai Jain, Shreyas Manjunath, Mrinal Sarmah, Samuel Hsu, Yaswanth Raparti, and Nitesh Pipralia. Weight block sparsity: Training, compilation, and ai engine accelerators, 2024. URL https://arxiv.org/abs/2407.09453.
> > >
> > > [3]Yulhwa Kim. cutlass_tilesparse. GitHub repository, 2023. Available at: https://github.com/YulhwaKim/cutlass_tilesparse.
> > >
> > > [4]Hugging Face. pytorch_block_sparse. 2023, GitHub, https://github.com/huggingface/pytorch_block_sparse.

---

### Review · Reviewer_DyU9 · 2024-12-17

**Summary Of Contributions:**

The article presents a novel approach to training large-scale machine learning models with block-wise sparse matrices, aiming to reduce computational and memory costs specially in the training process. The authors argue that current methods for training such models are inefficient as they start with dense models and only later apply sparsity, leading to high training costs. The proposed method leverages Kronecker product decomposition to factorize block-wise sparse matrices, enabling efficient training. The article claims significant reductions in training parameters and floating-point operations (FLOPs) without compromising performance.

**Audience:**

Yes

**Broader Impact Concerns:**

No.

**Claims And Evidence:**

Yes

**Requested Changes:**

Please see the weaknesses.

**Strengths And Weaknesses:**

Strengthes：
1. The paper introduces a new perspective on training block-wise sparse models by connecting them with Kronecker product decomposition, which is a unique contribution to the field.
2. The authors provide a solid theoretical basis for their method, demonstrating that every block-wise sparse matrix can be represented using their decomposition without significant performance loss.
3. Extensive empirical studies support the claims, showing substantial reductions in training parameters and FLOPs, in some cases by up to 97%, with minimal impact on accuracy.

Weaknesses:
1. The Kronecker product decomposition may introduce complexity in implementation, which could be a barrier for some practitioners. Especially, the reshape of input activations and the multiplication of inputs and Kronecker weights.
2. The success of the method in finding the optimal block size might depend heavily on initial conditions or the specific characteristics of the data, which is not extensively discussed. Generally, the accuracy in one turn or epoch is not a good feedback for determine the pattern.
3. The method of Kronecker product decomposition deduces the parameters and FLOPs significantly. however, the reduced calculation is similar to the low-bit quantization method, which contains low-bit multiplication and scales out of matrix multiplication. The performance of these simplified computaion methods will be reduced in contrast to general computation method, this is also verified in the experiments.

---

> ### Author Response · Authors · 2024-12-29
>
> 1. Compared with matrix multiplication, reshape operation is not computationally expensive. In particular, in pytorch, If reshape can return a view, it's very efficient. This happens when the new shape is compatible with the original tensor's memory layout. In this case, no data is copied, and the operation is essentially free. This is the case in our implementation.
> As we stated in the paper, using Kronecker product decomposition, Operation $Wx$ is turned into operation $B^T \check{x} A$ which is still a matrix multiplication. In the implementation, we are not calculating ($A \otimes B$).
>
> 2. In order to determine the best pattern, we run the gradient descent for optimization problem (7) for several iterations (e.g., in figure 3, we ran the gradient descent for 50 epochs and 200 epochs). As we stated in example 2, optimization problem (7) still is more efficient than trying each pattern separately using group LASSO (or any other baseline) and picking the best one.
>
>
> 3. In our paper, we aim to find the block-wise sparse models while making the training period more efficient. Although Low-bit quantization can help with reducing computation, it cannot find block-wise sparse models. Note that our method is not against the low-bit quantization. Low-bit quantization can be applied on top of our method to make our method even more efficient.

---

### Review · Reviewer_89D7 · 2024-12-21

**Summary Of Contributions:**

This work introduces a novel approach to training sparse deep learning models by focusing on block-wise sparsity from initialization. The authors propose decomposing the weight matrix using Kronecker products, representing it as (S ⊙ A) ⊗ B, where S, A, and B are smaller matrices. This decomposition allows the model to learn sparse patterns efficiently through S while maintaining a compact parameter representation from the start of training. By leveraging this structure, the method achieves sparsity without the typical dense-to-sparse training pipeline. Theoretically, the authors prove that this method have the same performance of a model trained by the group LASSO or pruning technique, highlighting the improvement of training sparsely from the beginning without sacrificing performance. Additionally, the flops during forward and backward further indicates the efficiency of the proposed methods. At last, from experiments with different model architectures on MNIST and CIFAR-100, the proposed method outperforms group LASSO, iteratively pruning, and other baselines.

**Audience:**

Yes

**Claims And Evidence:**

Yes

**Requested Changes:**

1. I would like the authors to further compare with other factorization methods for matrix decomposition to show the advantages of the proposed method.
2. The author should further separate the block sparsity part and Kronecker product part. Currently, it is still not clear how much each component contribute to the reduction of memory and computation during training and inference.
3. I would like to see more experiments on model size larger than 100M or even 1B. The current model size is too small, and I don't think it is very meaningful to use sparse training for these models.

**Strengths And Weaknesses:**

Strengths:
1. The work is novel as the first end-to-end block sparse training method.
2. Experiment results show its good performance in comparison with other block sparse training method.
3. The author provides theoretical analysis to show the effectiveness of proposed method.

Weaknesses:
1. In this work, the efficient training is achieved by the Kronecker products instead of block sparsity, which means that we can also reduce the parameters during training using dense A, and B with Kronecker products. From this view, I was wondering why only use Kronecker products cannot improve efficiency. Additionally, some other sparse training method like Monarch [1] also use different factorizations to represent the dense, full weight matrix. I will expect the authors to further explain the advantages of Kronecker products with these factorizations.
2. The experiments only include MNIST and CIFAR-100 with very small models. Therefore, with the increased model size these days, I am concerning about the performance on more modern models
3. This work do not explain why using block sparsity for sparse training. Instead, it directly focus on the new method from block sparse training.

Reference:
[1]. Dao, Tri, et al. "Monarch: Expressive structured matrices for efficient and accurate training." International Conference on Machine Learning. PMLR, 2022.

---

> ### Author Response · Authors · 2024-12-29
>
> 1. Kronecker product decomposition makes the training procedure more efficient. Using the decomposition in equation (3), If $S^{[l]}$ is a sparse matrix, then $W^{[l]}$ will be a blockwise sparse matrix. Therefore, during training, we find $S^{[l]}, A_i^{[l]}, B_i^{[l]}$ (this decreases the number of training parameters significantly). Then, during inference, we can use blockwise sparse matrix $W^{[l]} = \sum_{i=1}^{r_l} (S^{[l]}\odot A^{[l]}_i)\otimes B^{[l]}_i$ directly. On the other hand, Monarch is using several sparse matrices during training. The matrices have a specific sparsity pattern and several blocks set to zero before starting the training procedure.  After finishing training, the final weight matrix used in inference is dense.
> In our approach, we do not set any block to zero before training and we let the gradient descent find the block that should be zero. The final matrix during inference is a blockwise sparse model as well.
>
>
>
> 2. We repeat the experiment in table 3 for the ViT-base(86M) and ViT-Large(307M), here are the results: our method on ViT-base introduces 27.65M  training parameters and achieves 69.82% accuracy with 57.80% sparsity. On ViT-large, our method introduces 59.79M parameters and achieves 69.91% accuracy and 78.18% sparsity. We are running the baselines for ViT-base(86M) and ViT-Large(307M) and we will update table 3 as soon as we finish the experiment.
>
> 3. In the first paragraph of introduction, we mentioned the importance of block-wise sparsity for efficient inference. And our method enables us to find blockwise sparse matrices efficiently. We will also add one paragraph on importance of block-wise sparsity in related work section and will cite the following works:
>
> [1]Peng Xu, Wenqi Shao, Mengzhao Chen, Shitao Tang, Kaipeng Zhang, Peng Gao, Fengwei An, Yu Qiao, and Ping Luo. Besa: Pruning large language models with blockwise parameter-efficient sparsity allocation, 2024. URL https://arxiv.org/abs/2402.16880.
>
> [2]Scott Gray, Alec Radford, and Diederik P. Kingma. Gpu kernels for block-sparse weights. 2017. URL https://api.semanticscholar.org/CorpusID:52220661.
>
> [3] Paolo D’Alberto, Taehee Jeong, Akshai Jain, Shreyas Manjunath, Mrinal Sarmah, Samuel Hsu, Yaswanth Raparti, and Nitesh Pipralia. Weight block sparsity: Training, compilation, and ai engine accelerators, 2024. URL https://arxiv.org/abs/2407.09453.
>
> Request changes:
> 1. As we replied to weakness 1, we use equation (3), for efficient training. If $S^{[l]}$ is a sparse matrix, then $W^{[l]}$ will be a blockwise sparse matrix. During training, we find $S^{[l]}, A_i^{[l]}, B_i^{[l]}$ (this decreases the number of training parameters significantly). Then, during inference, we can use blockwise sparse matrix $W^{[l]} = \sum_{i=1}^{r_l} (S^{[l]}\odot A^{[l]}_i)\otimes B^{[l]}_i$ directly. Unfortunately, we cannot find block-wise sparse models using other types of matrix factorization.
>
> 2. The number of FLOPs and number of training parameters stated in papers are related to the training procedure. The sparsity rate is related to the inference time. We did not separate the Kronecker product from block-wise sparsity. As we mentioned, during training, we find $S^{[l]}, A_i^{[l]}, B_i^{[l]}$ (this decreases the number of training parameters and FLOPs significantly). During inference, we use $W^{[l]}$ which is a blockwise sparse matrix. Note that, the baselines also find block-wise sparse matrices and they are efficient during inference. However, they are not necessarily efficient during training. As a result, we would like to focus more on FLOPs and the number of parameters during training.
>
> In order to make the paper more readable, we replace word **FLOPs** with term **training FLOPs**, and we replace the term **sparsity rate** with **inference sparsity rate**.
>
> 3. As we replied to weakness 2, we are repeating the experiments for the ViT-base(86M) and ViT-Large(307M), and will update table 3 as soon as we finish the experiment.

---

> > ### Comment · Reviewer_89D7 · 2024-12-29
> > **Response from Reviewer 89D7**
> >
> > Dear authors,
> >
> > Thank you for your response!
> >
> > 1. For weakness 1, I am fully understanding the advantages of the proposed method now. Therefore, I think some modifications should be necessary for better understanding. During the training stage, the reduction of computation/memory cost during the initial stage of training should come from the matrix fraction, where S should be a dense matrix at that time. During inference, as we will directly use the weight W by combining all matrix fractions, the existence of S ensures W to be a block sparse matrix and accelerate inference. Therefore, I think clarifying that the efficiency comes from different components during training and inference in the edited paper will improve the readability and solve my problem, making the method stand out previous sparse/matrix fraction methods.
> >
> > 2. For weakness 2, adding more experiments on larger models will definitely help to validate the method. Additionally, I will recommend the authors to consider pre-training some small language models (i.e. 100M variant of Pythia [1]) as they are the popular topic in recent days.
> >
> > 3. A new problem that I think the paper should fix is that it don't measure the memory during training and inference for different methods. This is very important especially for the initial stage of training. For memory-constraint settings, we aim to keep a low memory requirement throughout training and inference, in which setting we should care about the cost for the initial stage. In contrast, the improvement in flops can accelerate training, but we can accept the dense initialization with a little more cost if the most of training is sparse. Based on the view, measuring and emphasizing the memory reduction is very important as it address an unsolved setting which is important. Additionally, the end-to-end training and inference time should also be provided. The Flop  itself do not indicate the practical efficiency improvement.
> >
> > 4. I guess that the calculation of matrix fractions may increase the training time. Therefore, as the method mainly aims to solve the memory and computation costs during the initial stage. I was wondering if we switch back to some other sparse method without matrix fraction, will this lead to a better training speed? (This is also the reason why we should know end-to-end improvement in latency and memory, as we need to see whether it is worth to use this method in the whole training)
> >
> > Reference:
> >
> > [1] Biderman, Stella, et al. "Pythia: A suite for analyzing large language models across training and scaling." International Conference on Machine Learning. PMLR, 2023.

---

> > > ### Author Response · Authors · 2025-01-02
> > >
> > > 1.Thanks for your comment, we will clarify the difference between training and inference in our revised version.
> > >
> > > 2.Thanks for your recommendation. We will include results on larger models including ViT-base(86M) and ViT-Large(307M) as well as 100M variant of Pythia in the paper. The experiments take a while and will upload the revised version as soon as the experiments are done.
> > >
> > > 3.During inference, since our method and baselines use sparse matrix  $\hat{W}^{[l]}$, the memory reduction for baselines and our method is approximately proportional to the sparsity rate (we say approximatly because the actual reduction depends on the implementation of sparse matrices in pytorch, tensorflow, or other deep learning libararies). During training, the memory required for model parameters is proportional to the number of training parameters. We have included both the sparsity rate and the number of training parameters in our experiments. We will clarify that the sparsity rate is equivalent to memory reduction during inference and training parameters is equivalent to memory usage during training.
> > >
> > > 4.We choose Kronecker product decomposition since block-wise sparsity can be achieved by using sparse mask S. On the other hand, it may not be possible to induce block-wise sparsity using other matrix decomposition methods (e.g. low-rank decomposition).

---

### Author Response · Authors · 2025-02-24
**Follow-up**

Dear Reviewers and Action Editor,

Could you please let us know if you need anything from us? We would be happy to update our manuscript if there is something that needs to be addressed. We submitted a revised version on January 10th and would appreciate any feedback.

Thank you!

---

### Decision · Action_Editor_PHWR · 2025-03-04

**Recommendation:** Accept as is

**Comment:**

The paper presents novel work, namely the first end-to-end block sparse training method.
Although it relies on known ingredients, the approach is overall novel.
 Experimental results show the good performance of the proposed approach in comparison with other block sparse training method. Finally, the author provides theoretical analysis to show the effectiveness of proposed method.

**Audience:**

The paper is interesting to a relatively wide audience, as it addresses the  improvement of learning with structured sparse priors.

**Claims And Evidence:**

The claims of the paper are well grounded and supported by clear evidence. Meaningful experiments are carried out to support the claims, and the authors convincingly get their main points through.